# Efficient Batched Algorithm for Contextual Linear Bandits with Large Action Space via Soft Elimination

**Osama A. Hanna**
University of California, Los Angeles
`ohanna@ucla.edu`

**Lin F. Yang**
University of California, Los Angeles
`linyang@ucla.edu`

**Christina Fragouli**
University of California, Los Angeles
`christina.fragouli@ucla.edu`

## Abstract

In this paper, we provide the first efficient batched algorithm for contextual linear bandits with large action spaces. Unlike existing batched algorithms that rely on action elimination, which are not implementable for large action sets, our algorithm only uses a linear optimization oracle over the action set to design the policy. The proposed algorithm achieves a regret upper bound $\tilde{O}(\sqrt{T})$ with high probability, and uses $O(\log \log T)$ batches, matching the lower bound on the number of batches [13]. When specialized to linear bandits, our algorithm can achieve a high probability gap-dependent regret bound of $\tilde{O}(1/\Delta_{\min})$ with the optimal $\log T$ number of batches, where $\Delta_{\min}$ is the minimum reward gap between a suboptimal arm and the optimal. Our result is achieved via a novel soft elimination approach, that entails "shaping" the action sets at each batch so that we can efficiently identify (near) optimal actions.

## 1 Introduction

In contextual linear bandits, a learner interacts with an environment over $T$ rounds: in each round $t$ the learner observes a (possibly different due to context change) set of actions $\mathcal{A}_t \subseteq \mathbb{R}^d$, plays one of them, and receives a reward that follows a noisy linear function parametrized by an unknown vector in $\mathbb{R}^d$. The objective of the learner is to minimize regret - how much reward it loses over the $T$ rounds by not always playing the "highest reward" (optimal) action. To achieve this, the learner at each round updates its policy (its method to select what action to play) based on what it has learned from all past actions played and rewards observed. Linear bandits form the special case where the action set is always the same, i.e., $\mathcal{A}_t = \mathcal{A}$ for all rounds $t$. Contextual linear and linear bandits have been widely investigated due to their significance in both theory and practice (eg., see [24]).

**Batched Setting.** In numerous real-world use cases, the learner may be restricted to change the policy a limited (small) number of times. This constraint may stem from factors such as computation or communication considerations, or may be imposed by the nature of the application, as is the case in multi-stage clinical trials or online marketing campaigns with high response rates, where it is not feasible to update the policy after each response. Similarly, the use of crowdsourcing platforms or the need to conduct time-consuming simulations in reinforcement learning may require policies with limited adaptivity. As a result, there has been significant interest in designing algorithms that can achieve the optimal regret with limited policy switches [30, 3, 31, 13, 12, 21, 32, 16]. This setup is known as the batched contextual linear bandit problem: the $T$ rounds are partitioned into batches, and the learner can collect rewards and update the action selection policy only at the end of each batch.

37th Conference on Neural Information Processing Systems (NeurIPS 2023).

**Large Action Space.** Contextual linear bandit applications frequently need to explore an extremely large (even continuous) set of actions, e.g., millions of products to be recommended. As other examples, in the classical bandit problem of clinical trials, each decision involves selecting a treatment option from a potentially infinite set of mixed treatments [15]. In manufacturing problems, the goal is often to maximize revenue by selecting from a very large set of decisions, with the revenue associated with each decision being unknown [36]. Additionally, in applications where actions correspond to images in a database or high-dimensional embeddings of complex documents like webpages, the set of actions can be vast [26, 5]. As a result, there is a strong interest in algorithms that can be efficiently implemented when the action space is large or infinite [11, 7, 19, 39, 41].

While computationally-efficient batched algorithms exist for contextual linear bandits with small action sets, and efficient ones that are not batched exist for contextual linear bandits with large action sets, to date, *there are no efficient batched algorithms* that can handle large action spaces. Existing batched algorithms for contextual linear bandits [32, 16] rely on *action elimination* that requires a linear scan of the action set; while efficient non-batched algorithms for large action spaces do not extend to the batched setting [32, 16] (see related work in the following for more details).

**Our Contributions.** In this paper, we provide the first efficient batched algorithm for contextual linear bandits with nearly optimal regret upper bound of $\tilde{O}(d^{3/2}\sqrt{T})$ with high probability, while using $O(\log \log T)$ batches, which matches the lower bound on the number of batches required to achieve $\sqrt{T}$-type regret bounds [13]. For linear bandits, our algorithm can attain a high probability gap-dependent regret bound of $\tilde{O}(d^3/\Delta_{\min})$ with the optimal $\log T$ number of batches [13], where $\Delta_{\min}$ represents the minimum reward gap between a suboptimal arm and the optimal.

Our algorithm for linear bandits, that we term SoftBatch, builds on a form of "soft elimination". Our observation is that, a good algorithm should be able to approximate the gap $\Delta(a)$ between each action $a \in \mathcal{A}$ and the optimal one with $O(\Delta(a))$ accuracy; and if we can do that, then we can use this knowledge to limit the number of times we play suboptimal actions, as well as use this knowledge to select which actions we want to play at all. As essentially all batched algorithms do, at each batch we select and play (a small number of) actions that enable to estimate well the unknown parameter vector without incurring large regret. In particular, for each batch, we choose a set of well-behaved basis actions (e.g., a barycentric spanner [7]), established by calling an optimization oracle polynomial times. However, instead of selecting at batch $m$, vectors from the "true" action set $\mathcal{A}$, we consider virtual "weighted" sets $\widetilde{\mathcal{A}}_m$, where each action's magnitude is weighted inversely proportional to the estimated gap $\Delta(a)$, and select vectors guided by these weighted action sets. Then we play each basis action $a$ a number of times inversely proportional to the square of the estimated gap $\Delta(a)$ to preserve small regret. This in return provides us an accurate estimator for the optimal parameter by the benign properties of the basis actions. Thus our approach implements a form of soft elimination (shaping) of the action set, where the actions closest to the optimal become increasingly dominant. A crucial part in our design is that we never actually calculate the gaps $\Delta(a)$ for all actions $a \in \mathcal{A}$ (only for the basis actions). The exploration policy we propose uses solely a linear optimization oracle applied to the original action set.

Our contextual bandit algorithm utilizes a recent reduction technique [18, 17] to transform the problem into a linear bandit problem. We incorporate the reduction into our batched linear bandit algorithm, by constructing an efficient linear optimization oracle for the exponentially large action set in the reduced problem using a linear optimization oracle for the original action sets (contexts).

Our proof techniques may be of independent interest. We develop a novel approach to bound regret in linear bandits, we design an efficient exploration policy using inverse squared gap weighting, and a simple method to handle the case where the action set does not span $\mathbb{R}^d$, where $d$ is the problem dimension. Our approach avoids the necessity of imposing assumptions, such as the one in [41], which entails having a subset of $d$ actions forming a matrix with determinant at least $r^d$ for a constant $r$. These assumptions can be strong, particularly when dealing with changing action sets, and may not hold after modifying the action set, for instance, by eliminating or weighting actions.

**Related Work.** Contextual linear and linear bandits have had significant impact both in theory and practice [1, 11, 28, 27, 24, 26, 37, 4, 8, 10]). The best performing algorithms achieve a regret bound $\tilde{O}(d\sqrt{T})$[1] [1, 11], matching the regret lower bound $\Omega(d\sqrt{T})$[24]. The same algorithms achieve a

---

[1] $\tilde{O}$ hides log factors.

nearly optimal regret upper bound $\tilde{O}(\frac{d^2}{\Delta_{\min}})$ if the minimum gap of suboptimal arms is lower bounded by $\Delta_{\min}$. However, the resulting policies require updates at every time step and involve solving a non-convex optimization problem, which is not practical for large action spaces [33, 11].

**Batched algorithms.** Existing batched algorithms for contextual linear bandits [32, 16, 18] have achieved nearly optimal regret upper bounds of $\tilde{O}(d\sqrt{T})$. However, these algorithms rely on action elimination, which involves either performing a linear scan on the action set or solving an optimization problem over the non-convex set of good (not eliminated) actions to design and implement the policy at each time step. Similarly, batched algorithms for linear bandits [25, 12] also rely on action elimination. Although, unlike contextual bandits, the elimination constraint in linear bandits can be linear, which can be exploited to efficiently compute the policy (under certain assumptions) [7], resulting in an $\tilde{O}(d^{3/2}\sqrt{T})$ regret upper bound, it requires solving an optimization problem over the action set with an elimination constraint. This can be much harder than solving the optimization problem over the action set without additional constraints for some sets, such as the non-convex set resulting from the reduction of contextual to linear bandits [18].

**Efficient algorithms for large action spaces.** There is a long line of work on efficient algorithms for linear bandits that only rely on a linear optimization oracle over the action set [7, 11, 9, 19, 20]. However, these algorithms cannot be extended to the batched setting without extra assumptions on the action set, and more importantly, they do not extend to the batched contextual setting. Existing efficient algorithms for contextual linear bandits [6, 41, 11] can achieve $\tilde{O}(d^{3/2}\sqrt{T})$ regret bound, but it remains unclear if they can be extended to the batched setting, particularly given the challenge posed by changing action sets. Another line of work attempts to design efficient algorithms using hashing-based methods to approximate the maximum inner product [39, 22], but these methods result in complexity that is sublinear but still polynomial in the number of actions.

Table 2 in App. A summarizes how our results position w.r.t. related work.

## 2 Model and Notation

**Notation.** We use $[n]$ for a natural number $n$ to denote the set $\{1, \cdots, n\}$; $\mathbf{1}(E)$, for an event $E$, to denote the indicator function which returns 1 if $E$ holds and 0 otherwise; $\mathcal{B}_r = \{a \in \mathbb{R}^d | \|a\|_2 \leq r\}$ to denote the ball of center 0 and radius $r$; $\mathcal{S}_r = \{a \in \mathbb{R}^d | \|a\|_2 = r\}$ to denote the sphere of center 0 and radius $r$; and $\|a\|_{\mathbf{V}} = \sqrt{a^\top \mathbf{V} a}$ to denote the matrix norm of a vector $a \in \mathbb{R}^d$ with respect to a positive semi-definite matrix $\mathbf{V}$. Table 1 in App. A summarizes our notation.

**Contextual Linear Bandits.** We consider a contextual linear bandit problem over an horizon of length $T$, where at each round $t \in [T]$, the learner receives a set of actions $\mathcal{A}_t \subseteq \mathbb{R}^d$ sampled from an unknown distribution $\mathcal{D}$ independently from other rounds. The learner plays an action $a_t \in \mathcal{A}_t$ and receives a reward $r_t = \langle a_t, \theta_\star \rangle + \eta_t$, where $\theta_\star$ is an unknown system parameter vector with $\theta_\star \in \mathbb{R}^d$, and $\eta_t$ is noise that is zero mean conditioned on the filtration of historic information $(\mathcal{A}_1, a_1, r_1, \cdots, \mathcal{A}_t, a_t)$. The learner adopts a **policy** that maps the history $(\mathcal{A}_1, a_1, r_1, \cdots, \mathcal{A}_t)$ to a distribution over the action set $\mathcal{A}_t$, with the objective of minimizing the pseudo regret defined as

$$R_T = \sum_{t=1}^{T} \sup_{a \in \mathcal{A}_t} \langle a - a_t, \theta_\star \rangle. \tag{1}$$

For simplicity, we assume that $\mathcal{A}_t$ is compact for all $t \in [T]$ almost surely, which ensures the existence of an action $a_\theta \in \mathcal{A}_t$ that attains the supremum $\sup_{a \in \mathcal{A}_t} \langle a, \theta \rangle$. Non-compact sets can be handled using sufficiently small approximations. We also adopt the following standard assumption [24].

**Assumption 1.** *(Boundedness.)* $\theta_\star \in \mathcal{B}_1$, $\mathcal{A}_t \subseteq \mathcal{B}_1$, and $|r_t| \leq 1$ almost surely $\forall t \in [T]$.

**Linear Bandits.** Changing the action set over time enables to model contextual information. If the action space is fixed, namely, $\mathcal{A}_t = \mathcal{A}$ for for all $t \in [T]$, the problem is known as Linear Bandits. For Linear Bandits, we denote an optimal action by $a^\star = \arg\max_{a \in \mathcal{A}} \langle a, \theta_\star \rangle$ and define the gap $\Delta_a = \langle a^\star - a, \theta_\star \rangle$ for all actions $a \in \mathcal{A}$.

**Batched Setting.** In a batched setting, the learner is only allowed to change the policy at $M$ pre-chosen rounds, where $M$ is the number of batches. Batch $m$ includes $T_m$ rounds, $m \in [M]$, with $\sum_{m=1}^{M} T_m = T$. In each batch, the learner adopts a policy $\pi$ that takes as input the action set $\mathcal{A}_t$ along with all the previous history except for rewards observed in the current batch, and outputs a

distribution over the action set $\mathcal{A}_t$. In particular, the rewards of the actions pulled in the current batch are utilized solely to update the policy at the end of the batch.

**Regularized least squares.** Let $\{a_i, r_i\}_{i=1}^n$ be a sequence of $n$ pulled actions and observed rewards over $n$ rounds. The regularized least squares estimate $\hat{\theta}$ of $\theta_\star$ based on this action-reward sequence can be calculated as

$$\hat{\theta} = \mathbf{V}^{-1} \sum_{i=1}^n r_i a_i, \tag{2}$$

where $\mathbf{V} = \lambda \mathbf{I} + \sum_{i=1}^n a_i a_i^\top$, and $\lambda$ is the regularization parameter.

**Goal.** Our goal is to design efficient batched algorithms for Contextual Linear and Linear Bandits with large (even infinite) action spaces that achieve (nearly) optimal regret.

We will do so by making use of the linear optimization oracles defined next.

**Definition 1.** A **linear optimization oracle** for a set $\mathcal{A}$ is a function $\mathcal{O}(\mathcal{A}; .)$ which takes as input $\theta \in \mathcal{B}_1$ and outputs $\mathcal{O}(\mathcal{A}; \theta) \in \mathcal{A}$ with $\langle \mathcal{O}(\mathcal{A}; \theta), \theta \rangle = \sup_{a \in \mathcal{A}} \langle a, \theta \rangle$. An **approximate linear optimization oracle with additive error** at most $\epsilon$ for the set $\mathcal{A}$ is a function $\mathcal{O}_\epsilon^+(\mathcal{A}; .) : \mathcal{B}_1 \to \mathcal{A}$ that satisfies $\langle \mathcal{O}_\epsilon^+(\mathcal{A}; \theta), \theta \rangle \geq \sup_{a \in \mathcal{A}} \langle a, \theta \rangle - \epsilon, \ \forall \theta \in \mathcal{B}_1$. An **approximate linear optimization oracle with multiplicative error** $0 < \alpha < 1$ for the set $\mathcal{A}$ is a function $\mathcal{O}_\alpha^\times(\mathcal{A}; .) : \mathcal{B}_1 \to \mathcal{A}$ that satisfies $\langle \mathcal{O}_\alpha^\times(\mathcal{A}; \theta), \theta \rangle \geq (1 - \alpha) \sup_{a \in \mathcal{A}} \langle a, \theta \rangle, \ \forall \theta \in \mathcal{B}_1$.

**Assumption 2.** *(Linear optimization oracle.) We assume that we can access a linear optimization oracle $\mathcal{O}(\mathcal{A}_t; .)$ for each set of actions $\mathcal{A}_t$ with running time at most $\mathcal{T}_{opt}$ and space complexity $\mathcal{M}_{opt}$.*

We note that assuming a linear optimization oracle over $\mathcal{A}_t$ is natural [7, 11, 9, 19, 20, 41] since even if the learner perfectly learns the unknown parameter vector $\theta_\star$, the learner still needs to solve $\sup_{a \in \mathcal{A}_t} \langle a, \theta_\star \rangle$ to minimize the regret in (1).

## 3 Efficient Soft Elimination Algorithm for Linear Bandits

In this section we propose and analyze an algorithm (which we call SoftBach and describe in Algorithm 1) for linear bandits, that is, when $\mathcal{A}_t = \mathcal{A}$.

### 3.1 Main Result

The following two theorems, proved in App. D and E, respectively, formally state that Algorithm 1 achieves (nearly) optimal regret using $M = \lceil \log \log T \rceil + 1$ batches with sample and time complexities polynomial in $d$ and linear in $T$. We provide the algorithm description in Section 3.2 and a proof outline in Section 3.3.

**Theorem 1.** *Consider a linear bandit instance with action set $\mathcal{A} \subseteq \mathbb{R}^d$ and horizon $T$. There exists a universal constant $C$ and a choice for the batch lengths such that Algorithm 1 finishes in at most $M = \lceil \log \log T \rceil + 1$ batches with regret bounded as*

$$R_T \leq C \gamma \sqrt{T} \log \log T \text{ with probability at least } 1 - \delta, \tag{3}$$

*where $\gamma = 8d\sqrt{C_L(\log(1/\delta) + \log T)}$, $C_L = e^8 d$ and $\delta$ is a parameter. Moreover, if $\forall a \in \mathcal{A}$ with $\Delta_a > 0$ we have $\Delta_a \geq \Delta_{\min}$, then there exists a choice of batch lengths so that Algorithm 1 finishes in at most $M = \log_4 T$ batches with regret bounded as*

$$R_T \leq C \frac{\gamma^2}{\Delta_{\min}} \log T \text{ with probability at least } 1 - \delta. \tag{4}$$

Our regret bounds achieve nearly optimal dependency on $T$, and match the best known regret bounds of $\tilde{O}(d^{3/2}\sqrt{T})$ for (unbatched) efficient contextual linear bandit algorithms [6, 41, 11], while losing a $\sqrt{d}$ factor when compared to the $\Omega(d\sqrt{T})$ lower bound [24]. This extra $\sqrt{d}$ factor is due to relying on the best known method to design a notion of spanner of the set of actions (as we explain in section 3.2) with radius $\sqrt{C_L} = O(\sqrt{d})$ using linear optimization oracles. Any future improvement that reduces the radius from $O(\sqrt{d})$ to $O(1)$ will immediately result in nearly optimal regret bounds for Algorithm 1. The following result upper bounds the time and space complexity.

---
**Algorithm 1 [SoftBatch]** A Batched Algorithm for Linear Bandits
---
1: Input: action set $\mathcal{A} \subseteq \mathbb{R}^d$, horizon $T$, number of batches $M$, batch lengths $\{T_m\}_{m=1}^M$, confidence parameter $\delta$.
2: Let $\mathcal{A}' = \mathcal{A} \cup \mathcal{B}_{1/T}$, $\quad C_L = e^8 d$, $\quad \gamma = 8d\sqrt{C_L(\log(1/\delta) + \log T)}$.
3: Initialize: $\theta_1 = 0$, $a_1^\star$ is a random action in $\mathcal{A}$, $\Delta_1(a) = 1 \ \forall a \in \mathcal{A}'$, and $T_0 = 1$.
4: **for** $m = 1 : M$ **do**
5: $\quad$ Calculate $\{a_1, \ldots, a_d\} = \mathrm{LWS}(\mathcal{A}', \eta_m = \sqrt{T_{m-1}}/(8\gamma), a_m^\star, \theta_m)$.
6: $\quad$ For the set $\{a_1, \ldots, a_d\}$ assign $\pi(i) = \frac{1}{d}$, $\forall i \in [d]$.
7: $\quad$ **for** $i = 1 : d$ **do**
8: $\quad\quad$ If $a_i \notin \mathcal{B}_{1/T}$, calculate $\Delta_m(a_i) = \langle a_m^\star - a_i, \theta_m \rangle$ and pull it $n_m(i) = \left\lceil \frac{\pi(i)T_m/8}{(1+\sqrt{T_{m-1}\Delta_m(a_i)/(8\gamma)})^2} \right\rceil$ times. **go to** step 10 if the number of pulls in the current batch reaches $T_m$. Terminate Algorithm 1 if the total number of pulls reaches $T$.
9: $\quad$ Pull action $a_0 = a_m^\star$ for $\max\{0, T_m - \sum_{i=1}^d n_m(i)\}$ times.
10: $\quad$ Compute the regularized (with $\lambda = 1$) least squares estimator $\mathbf{V}_m = \mathbf{I} + \sum_{i=1}^{T_m} \tilde{a}_i \tilde{a}_i^\top$ and $\theta_{m+1} = \mathbf{V}_m^{-1} \sum_{i=1}^{T_m} r_i \tilde{a}_i$, and $\tilde{a}_i$ is the action pulled in $i$-th round of the batch.
11: $\quad$ Update $a_{m+1}^\star = \mathcal{O}_{\frac{1}{T}}^+(\mathcal{A}; \theta_{m+1})$.
---

**Theorem 2.** *Algorithm 1 finishes in $\tilde{O}(Td^2 + d^4 M + \mathcal{T}_{opt}d^3 M)$ runtime and uses $\tilde{O}(d^2 + \mathcal{M}_{opt})$ memory, where $\mathcal{T}_{opt}, \mathcal{M}_{opt}$ are the time and space complexity of the linear optimization oracle.*

We observe that unlike algorithms that require a linear scan on the action set, our space and time complexities are polynomial in the parameters $d$, $T$, and $\mathcal{T}_{opt}$.

### 3.2 SoftBatch (Algorithm 1) Description

**Intuition.** The main intuition behind SoftBatch is that, we do not need to necessarily eliminate suboptimal actions; it suffices to be able to select and play a small set of unique actions $\mathcal{C}_m$ in each batch $m$, that allows to estimate increasingly well the parameter vector $\theta_*$ and the best action $a^\star$ while playing suboptimal actions for a small number of times. Our algorithm proposes a novel way to select such sets $\mathcal{C}_m$ efficiently, through a form of "action set shaping" that we will describe in this section. Additionally, to learn $\theta_*$ while achieving a (nearly) optimal regret, SoftBatch plays each action $a \in \mathcal{C}_m$ a number of times $\propto 1/\Delta_a^2$, where $\Delta_a = \langle a_\star - a, \theta_\star \rangle$ is the gap for action $a$ (i.e., we play the suboptimal actions in $\mathcal{C}_m$ for a small number of times so as not to accumulate regret). SoftBatch enables to estimate the gap $\Delta_a$ within a constant factor for any action $a$ (yet only does so for a limited number of actions in each batch), and essentially uses the gaps $\Delta_a$ as a guide on which actions to play and for how many rounds each.

**Steps.** SoftBatch (Algorithm 1) takes as input the action set $\mathcal{A} \subseteq \mathbb{R}^d$, the horizon $T$, the number of batches $M$, and the batch lengths $\{T_m\}_{m=1}^M$, and operates as follows[2].

In batch $m$, the algorithm starts with a current estimate of the parameter vector $\theta_\star$, which we call $\theta_m$, and an estimate of the optimal action $a^\star$ which we call $a_m^\star$; note that given these, we are able to estimate for any action $a \in \mathcal{A}$ the gap $\Delta_m(a) = \langle a_m^\star - a, \theta_m \rangle$ (but we will only do so for the actions the algorithm actually plays). The algorithm then calls LWS, a Linear Weighted Spanner subroutine (described in Algorithm 2), that it feeds with an augmented action space $\mathcal{A}' = \mathcal{A} \cup \mathcal{B}_{1/T}$ for reasons we will explain later. LWS selects $d$ actions $\mathcal{C}_m = \{a_1, \cdots, a_d\}$ to play in batch $m$ (note that some of these may belong to $\mathcal{B}_{1/T}$ and will in this case not be played). Each of these $d$ actions $a_i$ is pulled $n_m(i) \propto \frac{\pi(i)}{\Delta_m(a_i)^2}$ times, where $\pi(.)$ is a uniform exploration distribution with value $1/d$ for all the $d$ actions. We show in the proof of Theorem 1 that $\sum_{i=1}^d n_m(i) \leq T_m$, $\forall m \in [M]$, with high probability. To guarantee that the length of the batch is $T_m$, the algorithm pulls $a_m^\star$ for the remaining rounds, if needed. At the end of the batch, the algorithm updates its estimate $\theta_{m+1}$ of the unknown parameter vector using regularized least squares.

---
[2] We discuss how to select $M$ and $\{T_m\}$ in App. D.

The remaining core part of the algorithm to discuss is the subroutine LWS, and we do so next. We start by providing our reasoning behind the LWS design.

**The LWS Algorithm.** Recall that we want LWS at each batch $m$ to select $d$ vectors $\{a_i\} \subseteq \mathcal{A}'$ such that, by playing each $n_m(i)$ times, we can create a least-squares estimate $\theta_{m+1}$ of $\theta_*$ that allows an accurate estimate of the product $\langle a, \theta_\star \rangle$ for all $a \in \mathcal{A}$. It is well-known (see [24]) that the error in estimating $\langle a, \theta_\star \rangle$ is proportional to $\|a\|_{\mathbf{V}_m^{-1}}$, where $\mathbf{V}_m = \mathbf{I} + \sum_{i=1}^{T_m} a_i a_i^\top$ is the least squares matrix we used to estimate $\theta_{m+1}$. Thus, essentially we want LWS to select $d$ vectors $\{a_i\}$ that maintain a small $\|a\|_{\mathbf{V}_m^{-1}}$ for all actions $a \in \mathcal{A}$[3]. We can do so using what is called a G-optimal design [23].

**Definition 2.** (G-optimal design) For any set $\mathcal{A} \subseteq \mathbb{R}^d$, a subset $\mathcal{S} \subseteq \mathcal{A}$, together with a distribution $\pi$ over $\mathcal{S}$ is said to be a $C$**-approximate optimal design** for $\mathcal{A}$ if for any $a \in \mathcal{A}$

$$\|a\|_{\mathbf{V}_\pi^{-1}}^2 \leq Cd, \tag{5}$$

where $\mathbf{V}_\pi = \sum_{a_i \in \mathcal{S}} \pi(i) a_i a_i^\top$[4]. When $C = 1$ this is referred to as a **G-optimal design**.

Notice that if we were to play each action $a_i$ for $n\pi(i)$ times, then $\mathbf{V}_\pi$ would be (approximately) a normalized least squares matrix since $\mathbf{V}_\pi + \mathbf{I}/n = \mathbf{V}/n$, and hence, $\|a\|_{\mathbf{V}^{-1}}^2 \leq \|a\|_{\mathbf{V}_\pi^{-1}}^2/n$.

It is well-known that for any compact set, there exists a 1-approximate optimal design [23] with $|\mathcal{S}| = d$. However, computing an 1-approximate optimal design is NP-hard in general [14, 35], even for small action sets. Computing a 2-approximate optimal design can be done in polynomial time [38], but the complexity scales linearly with the size of the action set. Instead, we adopt an approach introduced in [7], which efficiently constructs an $O(\sqrt{d})$-approximate optimal design using only a linear optimization oracle. This relies on the concept of a barycentric spanner, which we define next.

**Definition 3.** (Barycentric spanner) For any set $\mathcal{A} \subseteq \mathbb{R}^d$, a subset $\mathcal{S} = \{a_1, \cdots, a_d\} \subseteq \mathcal{A}$ is said to be a $C$**-approximate barycentric spanner** for $\mathcal{A}$ if any $a \in \mathcal{A}$ can be expressed as a linear combination of vectors in $\mathcal{S}$ with coeficients in $[-C, C]$.

It is easy to see that a $C$**-approximate barycentric spanner** together with **a uniform distribution** $\pi(i) = 1/d$ results in a $C\sqrt{d}$-approximate optimal design [19, 41]. And importantly, a $C$-approximate barycentric spanner for a set $\mathcal{A}$ can be constructed using at most $O(d^2 \log_C d)$ calls of a linear optimization oracle over the set $\mathcal{A}$ [19].

However, this is still not sufficient for us. Even though we can efficiently construct a $C\sqrt{d}$-approximate optimal design for $\mathcal{A}$, we do not want to pull these arms according to a uniform distribution; we want to pull action $a_i$ with estimated gap $\Delta_m(a_i)$ for $n_m(i) = \lceil \pi(i)T_m/(1 + \sqrt{T_{m-1}}\Delta_m(a)/(8\gamma))^2 \rceil$ times to control the regret (which can be thought of as using a weighted distribution[5]). But if we do not use the uniform distribution, the resulting least squares matrix $\mathbf{V}_m$ may not satisfy that $\|a\|_{\mathbf{V}_m^{-1}}$ is sufficiently small for all actions $a$.

To account for this, instead of finding a $C$-approximate barycentric spanner for the set $\mathcal{A}$, at each batch $m$ we consider a **virtual action set** $\tilde{\mathcal{A}}_m$, which we define as

$$\tilde{\mathcal{A}}_m = \{\phi_m(a) | a \in \mathcal{A}\}, \phi_m(a) = \frac{a}{1 + \eta_m \Delta_m(a)}, \tag{6}$$

where $\eta_m = \sqrt{T_{m-1}}/(8\gamma)$ and find actions $\{a_1, \cdots, a_d\} \in \mathcal{A}$ such that $\{\phi_m(a_i)\}_{i=1}^d$ forms a $C$-approximate barycentric spanner for $\tilde{\mathcal{A}}_m$. The least squares matrix at batch $m$ can be bounded as

$$\mathbf{V}_m = \mathbf{I} + \sum_{i=0}^d \tilde{n}_m(i) a_i a_i^\top \geq \sum_{i=1}^d \frac{\pi(i)T_m/8}{(1 + \eta_m \Delta_m(a_i))^2} a_i a_i^\top = \sum_{i=1}^d \pi(i) \frac{T_m}{8} \phi_m(a_i) \phi_m(a_i)^\top \tag{7}$$

with high probability[6], where $\tilde{n}_m(i)$ is the number of times action $a_i$ is played in batch $m$ and $a_0 = a_m^\star$. That is, playing actions $\{a_1, \cdots, a_d\} \in \mathcal{A}$ for $n_m(i)$ times each, can equivalently

---

[3]Adding reward samples from the estimated best action $a_m^\star$ can only improve the least squares estimator.

[4]This summation assumes finiteness of the set $\mathcal{S}$ which suffices for our application.

[5]The technique of inverse gap weighting was employed in [2, 41], albeit with a different weighting approach using inverse gap instead of squared inverse gap, as utilized in our proposed schemes.

[6]We show in the proof of Theorem 1 that $\sum_{i=1}^d n_m(i) \leq T_m \forall m \in [M]$ with high probability, hence, all the required $n_m(i)$ action pulls can be finished within the batch.

---

**Algorithm 2** Linear Weighted Spanner (LWS) Algorithm

---

1: Input: set of actions $\mathcal{A}$, parameter $\eta$, estimated best action $\hat{a}$, estimated parameter $\hat{\theta}$.
2: Initialize: $\tilde{a}_i = e_i$, where $e_i$ is the $i$-th basis vector of dimension $d$. Let $\mathbf{A} = [\tilde{a}_1, \cdots, \tilde{a}_d]$.
3: Let $C = \exp(1)$, $\Delta(a) = \langle \hat{a} - a, \hat{\theta} \rangle$, $\phi(a) = a/(1 + \eta\Delta(a))$.
4: **for** $i = 1, \cdots, d$ **do**
5:     Find $\theta$ with $\langle \theta, \tilde{a} \rangle = \det(\tilde{a}, \mathbf{A}_{-i})$, $\forall \tilde{a} \in \mathbb{R}^d$.
6:     $a^+ = \text{LW-ArgMax}(\mathcal{A}; \frac{\theta}{\|\theta\|_2}, \eta, \hat{a}, \hat{\theta})$, $a^- = \text{LW-ArgMax}(\mathcal{A}; \frac{\theta}{\|\theta\|_2}, \eta, \hat{a}, -\hat{\theta})$.
7:     $a_i = \arg\max_{b \in \{a^+, a^-\}} |\langle \phi(b), \theta \rangle|$, $\tilde{a}_i = \phi(a_i)$.

8:
9: **for** $i = 1, \cdots, d$ **do**
10:     Find $\theta$ with $\langle \theta, \tilde{a} \rangle = \det(\tilde{a}, \mathbf{A}_{-i})$, $\forall \tilde{a} \in \mathbb{R}^d$.
11:     $a^+ = \text{LW-ArgMax}(\mathcal{A}; \frac{\theta}{\|\theta\|_2}, \eta, \hat{a}, \hat{\theta})$, $a^- = \text{LW-ArgMax}(\mathcal{A}; \frac{\theta}{\|\theta\|_2}, \eta, \hat{a}, -\hat{\theta})$.
12:     $a = \arg\max_{b \in \{a^+, a^-\}} |\langle \phi(b), \theta \rangle|$.
13:     **if** $|\det((\phi(a), \mathbf{A}_{-i}))| \geq C|\det(\mathbf{A})|$ **then**
14:         $a_i = a$, $\tilde{a}_i = \phi(a)$.
15:         **go to** line 8.
16: **Return:** $a_1, \cdots, a_d$.

---

be thought of as playing actions $\{\phi(a_1), \cdots, \phi(a_d)\} \in \tilde{\mathcal{A}}_m$ for $\pi(i)T_m$ times each; and since $\{\phi(a_1), \cdots, \phi(a_d)\}$ form an approximate optimal design (through a barycentric spanner) for the set $\tilde{\mathcal{A}}_m$, the resulting least squares matrix will lead to small $\|\phi_m(a)\|_{\mathbf{V}_m^{-1}}$ values. In our proofs we show that a small enough $\|\phi_m(a)\|_{\mathbf{V}_m^{-1}}$ implies $\|a\|_{\mathbf{V}_m^{-1}} = O(\Delta_a)$ as a result of the scaling in $\phi(a)$. We prove in Lemma 5 in App. D that this allows to estimate $\Delta_a$ within a constant factor, which is all we need.

Intuitively, the virtual set $\tilde{\mathcal{A}}_m$ weighs the actions inversely proportional to the estimated gap $\Delta_m(a)$ and batch length $\sqrt{T_{m-1}}$: the larger the gap and $T_{m-1}$, the smaller magnitude the corresponding action has; this implements a form of soft elimination (shaping) of the action set, where the actions closest to the optimal become increasingly dominant as the batch length increases while the remaining fade out to zero. As a result, as $m$ increases, the span of the optimal design focuses on the space where actions have small gaps, allowing to better distinguish among them.

To complete SoftBach (Algorithm 1), one last step is missing. LWS (Algorithm 2) follows standard steps (in Algorithm 2, see [7] for detailed explanation) to calculate the C-approximate barycentric spanner for $\tilde{\mathcal{A}}_m$. But to follow these steps, it requires the ability to solve the non-linear optimization problem $\sup_{a \in \mathcal{A}} \langle \phi_m(a), \theta \rangle$, since $\phi_m(a) = a/(1 + \sqrt{T_{m-1}}\Delta_m(a)/(8\gamma))$ is nonlinear in $a$. To do so, we will use[7] an approximate oracle with multiplicative error, that we term **LW-ArgMax** and describe next.

**LW-ArgMax Algorithm.** LW-ArgMax (Algorithm 3) constructs an approximate oracle with $(1 - \alpha)$-multiplicative error for the optimization $\sup_{a \in \mathcal{A}} \langle \phi_m(a), \theta \rangle$. This is sufficient: we show in Lemma 2 that Algorithm 2 can use LW-ArgMax to compute a $C/\alpha$-approximate barycentric spanner for $\tilde{\mathcal{A}}_m$.

Recall that, before providing the action set $\mathcal{A}$ to Algorithm 2, SoftBatch extends to $\mathcal{A}' = \mathcal{A} \cup \mathcal{B}_{1/T}$[8]. This guarantees that: $\mathcal{A}'$ spans $\mathbb{R}^d$ (required to find a barycentric spanner [7]), and $\sup_{a \in \mathcal{A}'} \langle a, \theta \rangle \geq 1/T$ for all $\theta$ with $\|\theta\|_2 = 1$ which implies that any approximate optimization oracle with additive error less than $1/(2T)$ has multiplicative error of at most $1/2$. The extension of the set $\mathcal{A}$ results in the barycentric spanner possibly containing points not in $\mathcal{A}$. However, we show that removing these points only affects $\sup_{a \in \mathcal{A}} \|a\|_{\mathbf{V}^{-1}}$ by a constant factor, since $\mathcal{B}_{1/T}$ has a small radius. Extending the

---

[7]A related problem was faced in [41], but with a different function hence, the resulting strategy does not apply in our case. Both [41] and our solution use the standard idea of line search, albeit with different steps and different number of iterations. The proof that our line search provides an approximate optimization oracle turns out to be much more involved than that of [41].

[8]The linear optimization problem $\max_{a \in \mathcal{A}'} \langle a, \theta \rangle$ can be solved by comparing $\max_{a \in \mathcal{A}} \langle a, \theta \rangle$ and $\max_{a \in \mathcal{B}_{1/T}} \langle a, \theta \rangle = 1/T$.

---

**Algorithm 3** LW-ArgMax Algorithm

---

1: Input: set of actions $\mathcal{A}$, $\theta \in \mathcal{S}_1$, parameter $\eta$, estimated best action $\hat{a}$, estimate $\hat{\theta}$, horizon T.
2: Let $\Delta(a) = \langle \hat{a} - a, \hat{\theta} \rangle$, $\phi(a) = a/(1 + \eta \Delta(a))$.
3: Let $W = 3 \log T$, $N = 36 W \log^2(T)$, $s = 1 - 1/6 \log T$, $\epsilon' = (1 - \exp(-1))/(12 T^{7 + 12 \log T})$.
4: Initialize $z = 2^W$.
5: **for** $i = 1, \cdots, N + 1$ **do**
6: $\quad \tilde{\theta} = (1 + 1/W) z \theta + z^{1 + 1/W} \eta \hat{\theta}$
7: $\quad a_i = \mathcal{O}^+_{\epsilon'}(\mathcal{A}; \tilde{\theta}/\|\tilde{\theta}\|_2)$.
8: $\quad z \leftarrow zs$.
9: **Return:** $\arg \max_{a \in \{a_i\}_{i=1}^N} \langle \phi(a), \theta \rangle$.

---

set $\mathcal{A}$ to $\mathcal{A}'$ also handles the case where the span of $\mathcal{A}$ is smaller than $\mathbb{R}^d$, that was typically handled in literature by constructing a basis of $\mathcal{A}$ which can be complicated for some sets.

LW-ArgMax then builds on the following observation (proved as part of the proof of Lemma 1):

$$\arg \max_{a \in \mathcal{A}'} \langle \phi(a), \theta \rangle (\langle a, \theta \rangle)^{1/W} = \arg \max_{a \in \mathcal{A}'} \sup_{z \geq 0} L_z(a), \tag{8}$$

where $L_z(a) = z \cdot (1 + 1/W) \cdot \langle a, \theta \rangle - z^{1 + 1/W}(1 + \eta \Delta(a))$ and $\Delta(a) = \sup_{b \in \mathcal{A}} \langle b - a, \theta_\star \rangle \, \forall a \in \mathcal{A}'$.

By choosing $W$ to be large enough, the left hand side of (8) becomes a good approximation for $\langle \phi(a), \theta \rangle$. For a fixed $z$, the supremum on the right hand side of (8) reduces to a linear optimization over the set $\mathcal{A}'$ (that we solve using an approximate linear optimization oracle). Although the optimal value of $z$ is not known, it can be bounded (see equation (25) in App. B); thus LW-ArgMax scans between upper and lower bounds on the optimal $z$ with a constant multiplicative step. The pseudo-code is provided in Algorithm 3.

### 3.3 Proof Outline for Theorem 1

We start by proving that LW-ArgMax is an approximate linear optimization oracle for the set $\tilde{\mathcal{A}}$ with $1 - \exp(-3)$ multiplicative error. The result is stated in Lemma 1 and proved in App. B.

**Lemma 1.** *Let $T \geq 3, \eta \in \mathbb{R}, \hat{a} \in \mathbb{R}^d, \hat{\theta} \in \mathcal{B}_T$ be given parameters, and $\mathcal{A}$ be a given set. Let $\Delta(a), \phi(a)$ denote $\Delta(a) = \langle \hat{a} - a, \hat{\theta} \rangle, \phi(a) = a/(1 + \eta \Delta(a))$. If $\mathcal{B}_{1/T} \subseteq \mathcal{A} \subseteq \mathcal{B}_1, |\eta| \leq T$ and $1/2 \leq 1 + \eta \Delta(a) \leq T^2, \forall a \in \mathcal{A}$, then for any $\theta \in \mathcal{S}_1$, LW-ArgMax outputs an element $a \in \mathcal{A}$ such that*

$$\langle \phi(a), \theta \rangle \geq \exp(-3) \sup_{b \in \{\phi(b') | b' \in \mathcal{A}\}} \langle b, \theta \rangle. \tag{9}$$

The conditions of Lemma 1 are easy to verify for all batches $m$; namely, $\mathcal{B}_{1/T} \subseteq \mathcal{A} \subseteq \mathcal{B}_1$ holds as we extend the set of actions by adding $\mathcal{B}_{1/T}$ before feeding it into Algorithm 3 and the condition $1/2 \leq 1 + \eta \Delta(a) \leq T^2, \forall a \in \mathcal{A}$ is proved in Theorem 1 for all the inputs fed into Algorithm 3.

Given the result of Lemma 1, we next show that Algorithm 2 finds a $C/\alpha$-approximate barycentric spanner of the set $\tilde{\mathcal{A}}_m, \forall m \in [M]$. This is done by slightly adapting the proof of Proposition 2.5 in [7] to work with approximate linear optimization oracles instead of exact oracles. The result is stated in the following theorem and the proof is provided in App. C for completeness.

**Lemma 2.** *Let $\eta \in \mathbb{R}, \hat{a} \in \mathbb{R}^d, \hat{\theta} \in \mathbb{R}^d$ be given parameters, and $\mathcal{A}$ be a given set. Let $\Delta(a), \phi(a)$ denote $\Delta(a) = \langle \hat{a} - a, \hat{\theta} \rangle, \phi(a) = a/(1 + \eta \Delta(a))$. Suppose that $\langle \phi(\text{LW-ArgMax}(\theta)), \theta \rangle \geq \alpha \sup_{a \in \mathcal{A}} \langle \phi(a), \theta \rangle$, then Algorithm 2 computes a $C/\alpha$-approximate barycentric spanner for the set $\tilde{\mathcal{A}} = \{\phi(a) | a \in \mathcal{A}\}$ with at most $O(d^2 \log_C(d/\alpha))$ calls to LW-ArgMax.*

To build our regret bounds, we essentially prove that a number of pulls of $\propto \frac{\pi(i)}{\Delta_m(a_i)^2}$ for action $a_i$ enables to estimate the gap $\Delta_m(a_i)$ within a constant factor of the real gap $\Delta_{a_i}$. To do so, we start by providing an error bound for estimating $\langle \phi_m(a), \theta_\star \rangle$ using standard sub-Gaussian concentration inequalities. Then, through mathematical induction, we extend this bound to the error of the action mean estimates $\langle a, \theta_m \rangle$. Intuitively, if we believe that $\langle a, \theta_m \rangle$ is a good estimate of $\langle a, \theta_\star \rangle$ for all

actions, which implies $\Delta_m(a)$ is a good estimate of $\Delta_a$ at batch $m$, then even though the scale in $\phi_{m+1}$ by $\Delta_m(a)$, this property will continue to hold at batch $m + 1$. The constants multiplying $\Delta_m(a)$ in $\phi_m$ are carefully designed to enable this. Finally, we show that the inverse squared gap weighting of the distribution enables to tightly upper bound the regret. $\square$

## 4 Algorithm for Contextual Linear Bandits

Our algorithm for contextual linear bandits is based on a technique proposed in [18], which reduces the contextual linear to a linear bandit setting. However, we cannot directly apply the reduction from [18] in Algorithm 1, as the reduction is not necessarily computationally efficient. Instead, we build a new algorithm (see Algorithm 4 in App. G) that incorporates reduction steps from [18] within Algorithm 1. One challenge we encounter is the introduction of a large, non-convex action set through the reduction process. To address this, we construct a linear optimization oracle over the new action set in order to ensure the efficiency of Algorithm 4. Additionally, the reduction requires estimating the expected value of a function (explained next and in App. G), and we carefully design the batch lengths to perform this estimation effectively. The following theorem describes our main result.

**Theorem 3.** *Consider a contextual linear bandit instance with $\mathcal{A}_t$ generated from an unknown distribution $\mathcal{D}$. There exists a universal constant $C$ and choice for batch lengths such that Algorithm 4 finishes in $O(\log \log T)$ batches with regret upper bounded as*

$$R_T \leq C\gamma\sqrt{T} \log \log T$$

*with probability at least $1 - \delta$, where $\gamma = 10\sqrt{C_L d(\log(16M/\delta) + 57d \log^2(6T))}$. Moreover, the running time and space complexity are $\tilde{O}(d^4 + \mathcal{T}_{opt}d^3 T)$, $\tilde{O}(d^2 + \mathcal{M}_{opt})$ respectively.*

We next briefly review the reduction and refer the reader to [18] for a detailed description. The basic idea in [18] is to establish a linear bandit action for each possible parameter vector $\theta$ of the contextual bandit instance. This is achieved through the use of the function $g : \mathbb{R}^d \to \mathbb{R}^d$, which computes the expected best action under the context distribution $\mathcal{D}$ with respect to the parameter $\theta$: $g(\theta) = \mathbb{E}_{\mathcal{A} \sim \mathcal{D}}[\mathcal{O}(\mathcal{A}; \theta)]$, where $\mathcal{O}$ is an optimization oracle (see Definition 1). A key insight, as stated in Theorem 1 of [18], is that if $a_t = \mathcal{O}(\mathcal{A}_t; \theta_t)$ for some $\theta_t \in \mathbb{R}^d$, then the reward generated by the contextual bandit instance can be expressed as $r_t = \langle g(\theta_t), \theta_\star \rangle + \eta'_t$, where $\eta'_t$ is noise with zero mean conditioned on the history. Consequently, the reward can be viewed as generated by pulling action $g(\theta_t)$ in a linear bandit instance with an action set $\mathcal{X} = \{g(\theta)|\theta \in \Theta\}$. Moreover, the same theorem demonstrates that if a linear bandit algorithm is employed to choose $g(\theta_t) \in \mathcal{X}$ at round $t$ and thus play action $a_t = \mathcal{O}(\mathcal{A}_t; \theta_t)$, then $|R_T - R_T^L| = \tilde{O}(\sqrt{T})$ with high probability, where $R_T^L = \sum_{t=1}^T \sup_{\theta \in \Theta} \langle g(\theta) - g(\theta_t), \theta_\star \rangle$ is the regret of the algorithm on the linear bandit instance.

To estimate $g$, which depends on the unknown context distribution $\mathcal{D}$, [18] proposes using contexts observed in previous batches. Specifically, the function $g$ is replaced by $g^{(m+1)}(\theta) = \frac{1}{|H_m|} \sum_{t \in H_m} \mathcal{O}(\mathcal{A}_t; \theta)$ for all $\theta \in \Theta'$, where $H_m$ is the set of indices for rounds in batch $m$ and $\Theta' = [\theta]_q | \theta \in \Theta$ is a discretization of $\Theta$, $[\theta]_q = q\lfloor \theta\sqrt{d}/q \rfloor/\sqrt{d}$ and $q$ is the discretization parameter. The action set at batch $m$ is correspondingly modified as $\mathcal{X}_m = \{g^{(m)}(\theta)|\theta \in \Theta'\}$. It is also shown in [18] that $g^{(m)}$ is a good estimate of $g$ for all $\theta \in \Theta'$ with high probability.

To leverage this reduction, we modify Algorithm 1 by adapting the action set in each batch based on the estimate of $g$, i.e., the set $\mathcal{X}_m$ (note that we do not need to explicitly calculate the sets $\mathcal{X}_m$). However, an issue arises where the estimate of $\theta_\star$ at batch $m$ depends on the approximate optimal design from batch $m - 1$, which employs the action set $\mathcal{X}_{m-1}$ estimated from the contexts of batch $m - 2$. In the proof of Theorem 3, we demonstrate that this leads to regret proportional to $T_m/\sqrt{T_{m-2}}$. If the batch lengths grow rapidly, significant regret may occur. To mitigate this, we reduce the growth rate of batch lengths by allowing them to increase only when $m$ is odd (a similar technique was employed in [18]). The pseudo-code is in Algorithm 4.

To implement Algorithm 4 efficiently we need: (i) an **approximate linear optimization oracle for the set $\mathcal{X}_m$** with additive error at most $\epsilon = (1 - \exp(-1))/(12T^{7+12 \log T})$: we show in Lemma 8 in App. F that $g^{(m)}([\theta]_q)$ can be used as our approximate oracle for $q \leq \epsilon/2$; and (ii) an **inverse of the function $g^{(m)}$** to find $\theta_t$ associated with $g^{(m)}(\theta_t)$ to play the action $a_t = \mathcal{O}(\mathcal{A}_t; \theta_t)$: we observe

that all actions played by our algorithms (Algorithm 1 and 4) are the output of the approximate optimization oracle for some $\theta$; namely, for Algorithm 4 any pulled action is of the form $g^{(m)}([\theta]_q)$ for some input to the approximate oracle $\theta$. Hence, the inversion of $g^{(m)}$ for the actions pulled by Algorithm 4 can be performed by storing $[\theta]_q$ whenever the action $g^{(m)}([\theta]_q)$ is stored. This increases both the space and time complexity only by a constant factor.

**Gap-dependent regret bounds for contextual linear bandits.** The main difficulty in extending the gap-dependent regret bounds in Theorem 1 to the contextual case is that a large minimum action gap in the original action sets $\mathcal{A}_t$ does not imply a large gap in the reduced action set $\mathcal{X}$. As a simple example consider $d = 1$, $\theta_\star = 1$, and two action sets $\mathcal{A}_1 = \{-1, 1\}$, and $\mathcal{A}_2 = \{-1\}$. At each iteration the learner receives the action set $\mathcal{A}_1$ with probability $p$ and $\mathcal{A}_2$ with probability $1 - p$ independently from other iterations. Recall that the action set in the reduced instance $\mathcal{X} = \{g(\theta) | \theta \in [-1, 1]\}$, where $g(\theta) = \mathbb{E}_{\mathcal{A} \sim \mathcal{D}}[\arg\max_{a \in \mathcal{A}} \langle a, \theta \rangle]$. For $\theta \geq 0$ we have that $g(\theta) = p(1) + (1 - p)(-1) = 2p - 1$, while for $\theta < 0$ we have $g(\theta) = p(-1) + (1 - p)(-1) = -1$. Then $\mathcal{X} = \{-1, 2p - 1\}$. Therefore, the suboptimality gap is $\Delta_{\min} = (2p - 1)(1) - (-1)(1) = 2p$ which can be arbitrarily small depending on $p$. Note that in the original contextual bandit instance, the minimum gap is at least 2 for both action sets.

While it may be possible to provide gap dependent regret bounds for our algorithm in the contextual case, this will require more sophisticated regret analysis that does not only rely on the reduced linear bandit instance.

**Numerical results.** In App. I, we provide a numerical example to compare the computational complexity of computing the exploration policy of our algorithm versus the complexity of computing the policy in [40].

## 5  Conclusion

In this paper, we proposed the first efficient batched algorithm for contextual linear bandits with large action spaces. Our algorithm achieves a high-probability regret upper bound of $\tilde{O}(\sqrt{T})$, uses $O(\log \log T)$ batches, and has a computational complexity that is linear in $T$ and polynomial in $d$.

## Acknowledgments

This work was partially supported by the NSF grants #2007714 and #2221871, the DARPA grant #HR00112190130, and by the Army Research Laboratory under the Cooperative Agreement W911NF-17-2-0196.

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
