| | |
|---|---|
| $T$ | time horizon |
| $\theta_\star$ | unknown parameter vector in $\mathbb{R}^d$ |
| $\mathcal{A}$ | action set if fixed over time |
| $d$ | dimension of actions and unknown parameter |
| $\mu_a$ | mean of arm $a$: $\langle a, \theta_\star \rangle$ |
| $a^\star$ | best action: $\arg\max_{a \in \mathcal{A}} \mu_a$ |
| $\Delta_a$ | gap: $\mu_{a^\star} - \mu_a$ |
| $\Delta_{\min}$ | minimum gap: $\inf_{a \in \mathcal{A}: \Delta_a > 0} \Delta_a$ |
| $\mathcal{O}(\mathcal{A}; .)$ | linear optimization oracle for the set $\mathcal{A}$ |
| $\mathcal{O}_\epsilon^+(\mathcal{A}; .)$ | approximate linear optimization oracle with additive error $\epsilon$ for the set $\mathcal{A}$ |
| $\mathcal{O}_\alpha^\times(\mathcal{A}; .)$ | approximate linear optimization oracle with multiplicative error $\alpha$ for the set $\mathcal{A}$ |
| $\mathcal{T}_{\text{opt}}$ | time complexity of optimization oracle $\mathcal{O}(\mathcal{A}; .)$ |
| $\mathcal{M}_{\text{opt}}$ | space complexity of optimization oracle $\mathcal{O}(\mathcal{A}; .)$ |
| $a_t$ | pulled action at time $t$ |
| $R_T$ | regret: $\sum_{t=1}^T \Delta_{a_t}$ |
| $\eta_t$ | noise at time $t$ |
| $r_t$ | reward at time $t$ |
| $M$ | number of batches |
| $T_m$ | length of batch $m$ |
| $H_m$ | set of time slots for batch $m$ |
| $\lambda$ | least squares regularization parameter |
| $\mathbf{V}_m$ | least squares matrix at batch $m$: $\lambda \mathbf{I} + \sum_{t \in H_m} a_t a_t^\top$ |
| $\theta_{m+1}$ | least squares estimate at end of batch $m$: $\mathbf{V}_m^{-1} \sum_{t \in H_m} r_t a_t$ |
| $a_m^\star$ | estimates best action at batch $m$: $\mathcal{O}_{1/T}^+(\mathcal{A}; \theta_m)$ |
| $\Delta_m(a)$ | estimated gap at batch $m$: $\langle a_m^\star - a, \theta_m \rangle$ |
| $C_L$ | approximate optimal design parameter: $e^8 d$ |
| $\gamma$ | $2d\sqrt{C_L(\log(1/\delta) + \log T)}$ |
| $\phi_m(a)$ | scaled action at batch $m$: $\frac{a}{1 + \sqrt{T_{m-1}\Delta_m(a)/(8\gamma)}}$ |
| $\mathcal{A}'$ | extended action set: $\mathcal{A}' = \mathcal{A} \cup \mathcal{B}_{1/T}$ |
| $\tilde{\mathcal{A}}_m$ | weighted action set: $\{\phi_m(a) | a \in \mathcal{A}'\}$ |
| $\delta$ | confidence parameter |
| $\mathcal{C}_m$ | set of size $d$ such that $\{\phi_m(a) | a \in \mathcal{C}_m\}$ is a barycentric spanner for $\tilde{\mathcal{A}}_m$ |
| $W$ | parameter: $3 \log T$ |
| $C$ | universal constant |
| $\mathcal{B}_r$ | $\{a \in \mathbb{R}^d | \|a\|_2 \leq r\}$ |
| $\mathcal{S}_r$ | $\{a \in \mathbb{R}^d | \|a\|_2 = r\}$ |
| $\|a\|_{\mathbf{V}}$ | $\sqrt{a^\top \mathbf{V} a}$ |
| $\mathbf{1}(E)$ | indicator function: returns 1 if $E$ holds and 0 otherwise |
| $[n], n \in \mathbb{N}$ | $\{1, \cdots, n\}$ |

Table 1: Table with notation for the linear bandit setting

## Supplementary Material of the Paper: Efficient Batched Algorithm for Contextual Linear Bandit with Large Action Space via Soft Action Elimination

## A   Tables: table of notations and comparison with related work

Our notation is collected in Table 1. Table 2 compares our results with state-of-the-art literature results as discussed in Section 1.

## B   Proof of Lemma 1: approximate inverse gap weighted optimization

**Lemma.** *Let $T \geq 3, \eta \in \mathbb{R}, \hat{a} \in \mathbb{R}^d, \hat{\theta} \in \mathcal{B}_T$ be given parameters, and $\mathcal{A}$ be a given set. Let $\Delta(a), \phi(a)$ denote $\Delta(a) = \langle \hat{a} - a, \hat{\theta} \rangle, \phi(a) = a/(1 + \eta\Delta(a))$. If $\mathcal{B}_{1/T} \subseteq \mathcal{A} \subseteq \mathcal{B}_1, |\eta| \leq T$ and*

| Algorithm | Regret Bound | Context | Efficient | Number of batches |
|:---:|:---:|:---:|:---:|:---:|
| [25, 12] | $\tilde{O}(d\sqrt{T})$ | ✗ | requires assumptions | $O(\log T)$ |
| [7, 11, 9, 19, 20] | $\tilde{O}(d\sqrt{T})$ | ✗ | ✓ | T |
| [1, 11] | $\tilde{O}(d\sqrt{T})$ | ✓ | ✗ | T |
| [32, 16, 18] | $\tilde{O}(d\sqrt{T})$ | ✓ | ✗ | $O(\log \log T)$ |
| [6, 41, 11] | $\tilde{O}(d^{3/2}\sqrt{T})$ | ✓ | ✓ | T |
| This paper | $\tilde{O}(d^{3/2}\sqrt{T})$ | ✓ | ✓ | $O(\log \log T)$ |

Table 2: Comparison with related work

$1/2 \leq 1 + \eta\Delta(a) \leq T^2$, $\forall a \in \mathcal{A}$, *then for any* $\theta \in \mathcal{S}_1$, *LW-ArgMax (Algorithm 3) outputs an element* $a \in \mathcal{A}$ *such that*

$$\langle \phi(a), \theta \rangle \geq \exp(-3) \sup_{b \in \{\phi(b')|b' \in \mathcal{A}\}} \langle b, \theta \rangle. \tag{10}$$

*Proof.* To simplify notations we define the modified gap as

$$\tilde{\Delta}(a) := 1 + \eta\Delta(a). \tag{11}$$

We also define the function $L_z : \mathcal{A} \to \mathbb{R}$ as

$$L_z(a) = (1 + 1/W)z\langle a, \theta \rangle - z^{1+1/W}\tilde{\Delta}(a), \tag{12}$$

where $W$ is a parameter and we have set it to $3\log T$. The main part of the proof shows that the optimizer of $L_z$ for some $z$ is an optimizer of $\langle \phi(a), \theta \rangle(\langle a, \theta \rangle)^{1/W}$, which, as we also prove, is a good approximation of $\langle \phi(a), \theta \rangle$ for $W = 3\log T$. Towards that we first aim to show $\sup_{a \in \mathcal{A}} \langle \phi(a), \theta \rangle(\langle a, \theta \rangle)^{1/W} = (W \sup_{a \in \mathcal{A}, z \geq 0} L_z(a))^{1/W}$. The following boundedness properties will be repeatedly used in the proof

$$|\langle a, \theta \rangle| \leq \|a\|_2 \|\theta\|_2 \leq 1 \tag{13}$$

and by assumption we have that the modified gap can be bounded as

$$1/2 \leq \tilde{\Delta}(a) \leq T^2. \tag{14}$$

We start by proving the following property about the function $L_z$.

**Claim 1.**
$$\sup_{z \geq 0} L_z(a) = \begin{cases} \frac{1}{W}(\langle \phi(a), \theta \rangle)^W \langle a, \theta \rangle & \text{if } \langle a, \theta \rangle \geq 0 \\ 0 & \text{otherwise.} \end{cases} \tag{15}$$

*Proof.* We notice the following fact about the function $L_z$. For any $a \in \mathcal{A}$ with $\langle a, \theta \rangle \geq 0$ we have that $L_z(a)$ is a concave function of $z$ for $z \geq 0$, hence, by setting the derivative to 0, we observe that

$$\sup_{z \geq 0} L_z(a) = \frac{1}{W}\left(\frac{\langle a, \theta \rangle}{\tilde{\Delta}(a)}\right)^W \langle a, \theta \rangle, \tag{16}$$

where the supremum is attained by

$$z_a = \left(\frac{\langle a, \theta \rangle}{\tilde{\Delta}(a)}\right)^W. \tag{17}$$

We also notice that for any $a \in \mathcal{A}$ with $\langle a, \theta \rangle < 0$, since for all $a \in \mathcal{A}$, $\tilde{\Delta}(a) \geq 0$ we have that

$$\sup_{z \geq 0} L_z(a) = 0, \tag{18}$$

where the supremum is attained by $z_a = 0$. The result follows by combining (16) and (18). ■

The following fact follows from Claim 1.

**Claim 2.**
$$\sup_{a \in \mathcal{A}} \langle \phi(a), \theta \rangle \langle a, \theta \rangle^{1/W} = (W \sup_{a \in \mathcal{A}, z \geq 0} L_z(a))^{1/W}. \tag{19}$$

*Proof.* We notice that since $\|\theta/T\|_2 = 1/T$ since $\|\theta\|_2 = 1$ by assumption. Then $\theta/T \in \mathcal{B}_{1/T} \subseteq \mathcal{A}$. We also have that $\langle \theta/T, \theta \rangle = 1/T > 0$, hence, from Claim 1 we have

$$\sup_{a \in \mathcal{A}:\langle a,\theta\rangle\geq 0} \sup_{z\geq 0} L_z(a) \geq \sup_{z\geq 0} L_z(\theta/T) \overset{(i)}{=} \frac{1}{W}\left(\frac{\langle\theta/T,\theta\rangle}{\tilde{\Delta}(\theta/T)}\right)^W \langle\theta/T,\theta\rangle \overset{(ii)}{\geq} \frac{1}{W}\left(\frac{1/T}{T^2}\right)^W /T > 0, \tag{20}$$

where $(i)$ follows from Claim 1 and $(ii)$ uses $\tilde{\Delta}(a) \leq T^2$. It follows from (15) that

$$\sup_{a \in \mathcal{A}, z\geq 0} L_z(a) = \max\left\{0, \sup_{a\in\mathcal{A}:\langle a,\theta\rangle\geq 0}\sup_{z\geq 0} L_z(a)\right\} = \sup_{a\in\mathcal{A}:\langle a,\theta\rangle\geq 0}\sup_{z\geq 0} L_z(a)$$

$$= \sup_{a\in\mathcal{A}:\langle a,\theta\rangle\geq 0} \frac{1}{W}\left(\frac{\langle a,\theta\rangle}{\tilde{\Delta}(a)}\right)^W \langle a,\theta\rangle. \tag{21}$$

Moreover we have that $\frac{1}{W}\left(\frac{\langle a,\theta\rangle}{\tilde{\Delta}(a)}\right)^W \langle a,\theta\rangle \leq 0$ whenever $\langle a,\theta\rangle \leq 0$. We can also see that

$$\sup_{a\in\mathcal{A}:\langle a,\theta\rangle\geq 0} \frac{1}{W}\left(\frac{\langle a,\theta\rangle}{\tilde{\Delta}(a)}\right)^W \langle a,\theta\rangle > 0 \tag{22}$$

by noticing that $\theta/T \in \mathcal{A}$ and has a positive objective value. Hence, we have that

$$\sup_{a\in\mathcal{A}, z\geq 0} L_z(a) = \sup_{a\in\mathcal{A}:\langle a,\theta\rangle\geq 0} \frac{1}{W}\left(\frac{\langle a,\theta\rangle}{\tilde{\Delta}(a)}\right)^W \langle a,\theta\rangle = \sup_{a\in\mathcal{A}} \frac{1}{W}\left(\frac{\langle a,\theta\rangle}{\tilde{\Delta}(a)}\right)^W \langle a,\theta\rangle. \tag{23}$$

It follows that
$$\sup_{a\in\mathcal{A}} \langle\phi(a),\theta\rangle\langle a,\theta\rangle^{1/W} = (W\sup_{a\in\mathcal{A},z\geq 0} L_z(a))^{1/W}. \tag{24}$$

∎

In the following we assume that $\sup_{a\in\mathcal{A}} \langle\phi(a),\theta\rangle\langle a,\theta\rangle^{1/W}$ is attained by some $b^\star \in \mathcal{A}$ and also that $\sup_{a\in\mathcal{A}} \langle\phi(a),\theta\rangle$ is attained by some $a^\star \in \mathcal{A}$. The proofs can be extended to the case where the supremums are not attained by using sufficiently small approximation.

The proof continues as following

- We show that the algorithm uses $z_i$ that is close to $z_{b^\star} = (\langle\phi(b^\star),\theta\rangle)^W$ (the optimizer of $L_z(b^\star)$ in (17)) in some iteration $i$.

- We show that the solution of $\sup_{a\in\mathcal{A}} L_{z_i}(a)$, namely $\tilde{a}$ satisfying $L_{z_i}(\tilde{a}) = \sup_{a\in\mathcal{A}} L_{z_i}(a)$, is an approximate optimizer of the function $\langle\phi(a),\theta\rangle\langle a,\theta\rangle^{1/W}$.

- We finally show that an approximate optimizer of $\langle\phi(a),\theta\rangle\langle a,\theta\rangle^{1/W}$ is also an approximate optimizer of $\langle\phi(a),\theta\rangle$.

Towards the first step, we start by finding an upper and lower bound on $z_{b^\star} = (\langle\phi(b^\star),\theta\rangle)^W$. From (13) and (14), we have that

$$2^W \geq z_{b^\star} = (\langle\phi(b^\star),\theta\rangle)^W \overset{(i)}{\geq} (\langle\phi(b^\star),\theta\rangle\langle b^\star,\theta\rangle^{1/W})^W$$

$$\overset{(ii)}{\geq} (\langle\phi(\theta/T),\theta\rangle\langle\theta/T,\theta\rangle^{1/W})^W$$

$$= \frac{1/T^{1+W}}{\tilde{\Delta}(\theta/T)} \geq \frac{1}{T^{3+W}}, \tag{25}$$

where $(i)$ follows from $|\langle b^\star, \theta \rangle| \leq 1$ and $\langle \phi(b^\star), \theta \rangle > 0$ (see (22)), and $(ii)$ follows by definition of $b^\star$ as the maximizer of $\langle \phi(a), \theta \rangle \langle a, \theta \rangle^{1/W}$ over the set $\mathcal{A}$ and the fact that $\theta/T \in \mathcal{B}_{1/T} \subseteq \mathcal{A}$.

Then, we find an upper and lower bound on the values of $z$ used by the algorithm. Recall that Algorithm 3 starts with $z = 2^W$ where $W = 3 \log T$ and decreases $z$ with a factor of $s = 1 - \frac{1}{6 \log T}$ for $N = 36 W \log^2 T$ iterations. We have that

$$2^W s^N = 2^W (1 - \frac{1}{6 \log T})^N \leq \exp(W - N/(6 \log T)) = \exp(W(1 - 6 \log T))$$

$$\leq \exp(-3W \log(T)) = \frac{1}{T^{3W}} \leq \frac{1}{T^{3+W}} \qquad (26)$$

From (25), (26), the fact that Algorithm 3 starts with $z = 2^W$ and decreases $z$ by a factor of $s = 1 - \frac{1}{6 \log T}$ each step, it follows that there is an iteration $i$ with

$$s z_{b^\star} \leq z_i \leq z_{b^\star}, \qquad (27)$$

where $z_i$ is the value of the variable $z$ in iteration $i$. Now we consider the function $L_{z_i}$. We aim to show that an approximate optimizer of $L_{z_i}$ is an approximate optimizer of $\langle \phi(a), \theta \rangle \langle a, \theta \rangle^{1/W}$. This is proved in the following lemma.

**Lemma 3.** *Consider given* $\eta \in \mathbb{R}, \theta \in \mathbb{R}^d, \hat{\theta} \in \mathbb{R}^d, \hat{a} \in \mathbb{R}^d$ *and let* $W = 3 \log T, \Delta(a) = \langle \hat{a} - a, \hat{\theta} \rangle, \tilde{\Delta}(a) = 1 + \eta \Delta(a), \phi(a) = a/\tilde{\Delta}(a), L_z(a) = (1 + 1/W)z\langle a, \theta \rangle - z^{1+1/W}\tilde{\Delta}(a).$ *Let* $i$ *be an iteration of Algorithm 3 with* $s z_{b^\star} \leq z_i \leq z_{b^\star}$. *If* $\mathcal{B}_{1/T} \subseteq \mathcal{A} \subseteq \mathcal{B}_1$ *and* $1/2 \leq \tilde{\Delta}(a) \leq T^2 \ \forall a \in \mathcal{A}$, *then we have that*

$$\langle \phi(a_i), \theta \rangle \langle a_i, \theta \rangle^{1/W} \geq \exp(-1) \sup_{b \in \mathcal{A}} \langle \phi(b), \theta \rangle \langle b, \theta \rangle^{1/W}, \qquad (28)$$

*where $a_i$ is the approximate optimizer defined in step 7 of Algorithm 3 at iteration $i$.*

*Proof.* To utilize Claim 1 to relate the optimizer of $L_{z_i}$ to the optimizer of $\langle \phi(a), \theta \rangle \langle a, \theta \rangle^{1/W}$, we first show that $\sup_{a \in \mathcal{A}} L_{z_i}(a) > 0$. We have that (recall that $b^\star$ is the optimizer of $\langle \phi(a), \theta \rangle \langle a, \theta \rangle^{1/W}$)

$$L_{z_i}(b^\star) = (1 + 1/W)z_i\langle b^\star, \theta \rangle - z_i^{1+1/W}\tilde{\Delta}(b^\star)$$

$$\overset{(i)}{\geq} (1 + 1/W)s z_{b^\star}\langle b^\star, \theta \rangle - z_{b^\star}^{1+1/W}\tilde{\Delta}(b^\star)$$

$$\overset{(ii)}{=} (\langle \phi(b^\star), \theta \rangle \langle b^\star, \theta \rangle^{1/W})^W ((1 + 1/W)s - 1), \qquad (29)$$

where $(i)$ follows from $\langle b^\star, \theta \rangle > 0$ (see (22)) and $\tilde{\Delta}(b^\star) \geq 0$, and $(ii)$ follows by substituting $z_{b^\star} = (\langle b^\star, \theta \rangle / \tilde{\Delta}(b^\star))^W$. We denote

$$\beta := (1 + 1/W)s - 1. \qquad (30)$$

We next lower bound $\beta$ as follows (recall that $T \geq 3$ and $s = 1 - 1/6\log T$)

$$(W\beta)^{1/W} = (1/2 - 1/(6 \log T))^{1/(3 \log T)} \geq (1/4)^{1/(3 \log T)} \geq \exp(-0.5/\log T) \geq \exp(-0.5). \qquad (31)$$

It follows that

$$\beta \geq \exp(-0.5W)/(W) \geq 1/(3T^2), \qquad (32)$$

where the last inequality uses $T \geq 3$, hence, $\log T \leq \sqrt{T}$. Substituting in (29) we get that

$$\sup_{b \in \mathcal{A}} L_{z_i}(b) \geq L_{z_i}(b^\star) \geq \frac{(\langle \phi(b^\star), \theta \rangle \langle b^\star, \theta \rangle^{1/W})^W}{3T^2} \geq \frac{1}{3T^{2+12 \log T}}, \qquad (33)$$

where the last inequality follows by definition of $b^\star$ as the maximizer of $\langle \phi(a), \theta \rangle \langle a, \theta \rangle^{1/W}$ over the set $\mathcal{A}$ and the fact that $\theta/T \in \mathcal{B}_{1/T} \subseteq \mathcal{A}$. In the algorithm, we do not construct an optimizer for $L_{z_i}$; instead we use an approximate optimizer $a_i$ of the linear function given in step 7 of Algorithm 3. In the following we will use (33) to show that $L_{z_i}(a_i) > 0$. We notice that

$$L_{z_i}(b) = (1 + 1/W)z_i\langle b, \theta \rangle - z_i^{1+1/W}(1 + \eta\langle \hat{a} - b, \hat{\theta} \rangle)$$

$$= \langle b, (1+1/W)z_i\theta + z_i^{1+1/W}\eta\hat{\theta}\rangle - z_i^{1+1/W}(1+\eta\langle\hat{a},\hat{\theta}\rangle)$$
$$= \langle b, \tilde{\theta}_i\rangle - z_i^{1+1/W}(1+\eta\langle\hat{a},\hat{\theta}\rangle), \tag{34}$$

where $\tilde{\theta}_i = (1+1/W)z_i\theta + z_i^{1+1/W}\eta\hat{\theta}$. It follows that $\sup_{b\in\mathcal{A}} L_{z_i}(b) = (\sup_{b\in\mathcal{A}}\langle b,\tilde{\theta}_i\rangle) - z_i^{1+1/W}(1+\eta\langle\hat{a},\hat{\theta}\rangle)$. Hence, by definition of $a_i$ in Algorithm 3 we get that

$$L_{z_i}(a_i) \geq \sup_{b\in\mathcal{A}} L_{z_i}(b) - \frac{1-\exp(-1)}{12T^{7+12\log T}}\|\tilde{\theta}_i\|_2$$

$$\overset{(i)}{\geq} \sup_{b\in\mathcal{A}} L_{z_i}(b) - \frac{1-\exp(-1)}{3T^{2+12\log T}}$$

$$\overset{(ii)}{\geq} \sup_{b\in\mathcal{A}} L_{z_i}(b) - (1-\exp(-1))\sup_{b\in\mathcal{A}} L_{z_i}(b)$$

$$= \exp(-1)\sup_{b\in\mathcal{A}} L_{z_i}(b), \tag{35}$$

where $(i)$ follows from $\|\tilde{\theta}_i\|_2 \leq (1+1/W)z_i\|\theta\|_2 + |\eta|z_i^{1+1/W}\|\hat{\theta}\|_2 \leq 2T^2 2^{W+1} \leq 4T^5$ (recall that $|\eta|\leq T, \hat{\theta}\in\mathcal{B}_T$ and $z_i\leq 2^W$, $W=3\log T$) and $(ii)$ follows from (33). It follows from (33) that $L_{z_i}(a_i) > 0$. Hence, from (15) we get that $\langle a_i,\theta\rangle \geq 0$. From (15) again it follows that

$$\frac{1}{W}\left(\frac{\langle a_i,\theta\rangle}{\tilde{\Delta}(a_i)}\right)^W\langle a_i,\theta\rangle = \sup_{z\geq 0} L_z(a_i) \geq L_{z_i}(a_i)$$

$$\overset{(i)}{\geq} \exp(-1)\sup_{b\in\mathcal{A}} L_{z_i}(b)$$

$$\geq \exp(-1)L_{z_i}(b^\star) \overset{(ii)}{\geq} \exp(-1)(\langle\phi(b^\star),\theta\rangle\langle b^\star,\theta\rangle^{1/W})^W\beta \tag{36}$$

where $(i)$ follows from (35) and $(ii)$ follows from (29). Hence, we have that

$$\langle\phi(a_i),\theta\rangle\langle a_i,\theta\rangle^{1/W} \geq \exp(-1/W)\langle\phi(b^\star),\theta\rangle\langle b^\star,\theta\rangle^{1/W}(W\beta)^{1/W}$$

$$\geq \exp(-0.5)\langle\phi(b^\star),\theta\rangle\langle b^\star,\theta\rangle^{1/W}(W\beta)^{1/W}$$

$$\overset{(i)}{\geq} \exp(-1)\langle\phi(b^\star),\theta\rangle\langle b^\star,\theta\rangle^{1/W}, \tag{37}$$

where $(i)$ follows from (31). $\blacksquare$

The last part of the proof shows that an approximate optimizer for $\langle\phi(a),\theta\rangle\langle a,\theta\rangle^{1/W}$ is also an approximate optimizer for $\langle\phi(a),\theta\rangle$. We lower bound $\langle\phi(a_i),\theta\rangle$ as follows (recall that $a^\star$ is the optimizer of $\langle\phi(a),\theta\rangle$)

$$\frac{\langle\phi(a_i),\theta\rangle}{\langle\phi(a^\star),\theta\rangle} = \frac{\langle\phi(a_i),\theta\rangle\langle a_i,\theta\rangle^{1/W}}{\langle\phi(a^\star),\theta\rangle\langle a^\star,\theta\rangle^{1/W}}(\frac{\langle a^\star,\theta\rangle}{\langle a_i,\theta\rangle})^{1/W}$$

$$\overset{(i)}{\geq} \exp(-1)\frac{\langle\phi(b^\star),\theta\rangle\langle b^\star,\theta\rangle^{1/W}}{\langle\phi(a^\star),\theta\rangle\langle a^\star,\theta\rangle^{1/W}}(\frac{\langle a^\star,\theta\rangle}{\langle a_i,\theta\rangle})^{1/W}$$

$$\overset{(ii)}{\geq} \exp(-1)(\frac{\langle a^\star,\theta\rangle}{\langle a_i,\theta\rangle})^{1/W}$$

$$\overset{(iii)}{\geq} \exp(-1)\langle a^\star,\theta\rangle^{1/W} = \exp(-1)\langle\phi(a^\star),\theta\rangle^{1/W}\tilde{\Delta}(a^\star)^{1/W}$$

$$\overset{(iv)}{\geq} \exp(-1)\langle\phi(a^\star),\theta\rangle^{1/W}0.5^{1/W}$$

$$\geq \exp(-1.5)\langle\phi(a^\star),\theta\rangle^{1/W}$$

$$\overset{(v)}{\geq} \exp(-1.5)\langle\phi(\theta/T),\theta\rangle^{1/W} = \exp(-1.5)(\frac{1/T}{\tilde{\Delta}(\theta/T)})^{1/W}$$

$$\overset{(vi)}{\geq} \exp(-1.5) \left(\frac{1/T}{T^2}\right)^{1/W} = \exp(-1.5 - 3\log T/W) = \exp(-2.5). \quad (38)$$

where $(i)$ follows from Lemma 3, $(ii)$ follows by definition of $b^\star$ as the maximizer of $\langle\phi(a),\theta\rangle\langle a,\theta\rangle^{1/W}$, $(iii)$ follows from $\langle a^\star,\theta\rangle > 0, \langle a_i,\theta\rangle > 0$ and $|\langle a_i,\theta\rangle| \leq 1$, $(iv)$ follows from (14), $(v)$ uses the fact that $\theta/T \in \mathcal{A}$ and definition of $a^\star$ to attain the supremum of $\langle\phi(a),\theta\rangle$, and $(vi)$ follows from (14). The proof is concluded by noticing that $a_i$ is one of the candidates in the return statement of Algorithm 3, hence, if $a$ is the output of Algorithm 3, then $\langle\phi(a),\theta\rangle \geq \langle\phi(a_i),\theta\rangle \geq \exp(-3)\langle\phi(a^\star),\theta\rangle$, where the last inequality follows from (38). ∎

## C  Proof of Lemma 2: barycentric spanner

We here prove that Algorithm 2 can efficiently find a barycentric spanner.

**Lemma 2.** *Let $\eta \in \mathbb{R}, \hat{a} \in \mathbb{R}^d, \hat{\theta} \in \mathbb{R}^d$ be given parameters, and $\mathcal{A}$ be a given set. Let $\Delta(a), \phi(a)$ denote $\Delta(a) = \langle\hat{a} - a, \hat{\theta}\rangle, \phi(a) = a/(1 + \eta\Delta(a))$. Suppose that for any $\theta \in \mathcal{S}_1$, LW-ArgMax (Algorithm 3) with inputs $\mathcal{A}, \theta, \eta, \hat{a}, \hat{\theta}$, outputs $a_\theta \in \mathcal{A}$ with $\langle\phi(a_\theta),\theta\rangle \geq \alpha \sup_{a\in\mathcal{A}}\langle\phi(a),\theta\rangle$, then Algorithm 2 computes a $C/\alpha$-approximate barycentric spanner for the set $\tilde{\mathcal{A}} = \{\phi(a)|a \in \mathcal{A}\}$ with at most $O(d^2 \log_C(d/\alpha))$ calls to LW-ArgMax.*

*Proof.* The proof is a simple modification of the proof of Proposition 2.5 in [7]; the difference is that we replace exact linear optimization oracles with approximate ones, and show that the resulting vectors still have the good properties we want.

We note that Lemma 2 holds for any generic action set $\mathcal{A}$ used to call Algorithm 2; however, since Algorithm 1 calls Algorithm 2 (and Algorithm 3) with input action set $\mathcal{A}'$, for consistency we will use $\mathcal{A}'$ as the input action set in the following, e.g., we interpret the lemma statement assumption as:

$$\langle\phi(\text{LW-ArgMax}(\theta)),\theta\rangle \geq \alpha \sup_{a\in\mathcal{A}'}\langle\phi(a),\theta\rangle.$$

From this asumption and the fact that $\max_{a\in\tilde{\mathcal{A}}}|\langle a,\theta\rangle| = \max\{\max_{a\in\tilde{\mathcal{A}}}\langle a,\theta\rangle, \max_{a\in\tilde{\mathcal{A}}}\langle a,-\theta\rangle\}$, we have that step 7 (and similarly step 12) in Algorithm 2 outputs $a$ with

$$|\langle\phi(a),\theta\rangle| \geq \alpha \sup_{\tilde{a}\in\tilde{\mathcal{A}}}|\langle\tilde{a},\theta\rangle|, \text{ for some } 0 < \alpha < 1. \quad (39)$$

We next show that if Algorithm 2 terminates then $\{\phi(a_1),\cdots,\phi(a_d)\}$ is a $C/\alpha$-approximate barycentric spanner. We have that if there exists $a' \in \mathcal{A}'$ with $|\det((\phi(a'),\mathbf{A}_{-i}))| \geq C/\alpha|\det(\mathbf{A})|$ for some $i$, then from (39), in step 12 we have an $a$ with $|\det((\phi(a),\mathbf{A}_{-i}))| \geq C|\det(\mathbf{A})|$, hence, the algorithm will continue. As a result when Algorithm 2 terminates we have that

$$\sup_{a\in\tilde{\mathbf{A}}}|\det((a,\mathbf{A}_{-i}))| \leq C/\alpha|\det(\mathbf{A})|, \quad \forall i \in [d]. \quad (40)$$

In the proof of Lemma 1 we showed that $\sup_{a\in\tilde{\mathcal{A}}}\langle a,\theta\rangle > 0, \forall\theta \neq 0$. This shows that at every step of Algorithm 2, the matrix $\mathbf{A}$ has non-zero determinent. Hence, $\{a_1,\cdots,a_d\}$ span $\mathbb{R}^d$. As a result for any $\tilde{a} \in \tilde{\mathcal{A}}$ we have that $\tilde{a} = \sum_{i=1}^d w_i a_i$ for some $\{w_i\}_{i=1}^d$. We have that

$$|\det(\tilde{a},\mathbf{A}_{-i})| = |\det(\sum_{i=1}^d w_i a_i, \mathbf{A}_{-i})| = |w_i||\det(\mathbf{A})|. \quad (41)$$

Hence, from (40) we get that

$$|w_i| \leq C/\alpha. \quad (42)$$

This implies that $\{\phi(a_1),\cdots,\phi(a_d)\}$ is a $C/\alpha$-approximate barycentric spanner for $\tilde{\mathcal{A}}$. It remains to show that Algorithm 2 terminates in $O(d^2 \log_C d)$ iterations. The number of iterations of the first for loop is $d$. To bound the number of iterations of the second for loop, we notice that for each repetition of the for loop (which takes at most $d$ iterations), $\det(\mathbf{A})$ increases by a factor of $C$. Let $\mathbf{M}_i = [\tilde{a}_1,\cdots,\tilde{a}_i,e_{i+1},\cdots,e_d]$ be the value of the matrix $\mathbf{A}$ at the end of the $i$-th iteration of the first for loop. As the determinant of $\mathbf{A}$ increases by at least factor of $C$ each repetition, then if $N$ is the number of repetitions of the second for loop, we have that $C^N \leq |\det(\mathbf{A})/\det(\mathbf{M}_d)|$, where $\mathbf{A}$ is

the matrix at the end of the $N$-th repetition of the second for loop. Hence, to prove the theorem it suffices to show that $|\det(\mathbf{A})/\det(\mathbf{M}_d)| \leq (1/\alpha)^d d^{d/2}$. Let $u_i^T = e_i^\top \mathbf{M}_i^{-1}$ and define $\mathbf{U}$ to be the matrix whose $i$-th row is $u_i$. We observe that

$$\langle u_i, a \rangle = \frac{\det(a, \mathbf{M}'_{-i})}{\det(\mathbf{M}_i)}, \quad \forall a \in \tilde{\mathcal{A}}, \tag{43}$$

by noticing that both sides are linear functions of $a$ and equality holds for all columns of $\mathbf{M}_i$ which form a basis for $\mathbb{R}^d$. It follows from (39) that $|u_i^\top a| \leq 1/\alpha$. As each entry of $\mathbf{UA}$ is $u_i^\top a$ for some $i \in [d], a \in \tilde{\mathcal{A}}$, all the entries of $\mathbf{UA}$ lie in $[-1/\alpha, 1/\alpha]$. Hence, $\det(\mathbf{UA}) \leq (1/\alpha)^d d^{d/2}$ as the determinant of a matrix is upper bounded by the product of the $L^2$-norms of its columns. We also notice that if $\mathbf{M}_d = [\tilde{a}_1, \cdots, \tilde{a}_d]$, then by definition of $u_i$ we have $\langle u_i, \tilde{a}_j \rangle$ is zero if $j < i$, and $\langle u_i, \tilde{a}_i \rangle = 1, \forall i \in [d]$. Hence, $\mathbf{UM}_d$ is upper triangular matrix with unit diagonal, implying $\det(\mathbf{UM}_d) = 1$. We have that

$$\frac{\det(\mathbf{A})}{\det(\mathbf{M}_d)} = \frac{\det(\mathbf{UA})}{\det(\mathbf{UM}_d)} \leq (1/\alpha)^d d^{d/2}. \tag{44}$$

This conlcudes the proof. ∎

## D Proof of Theorem 1: regret analysis for linear bandits

**Theorem 1.** *Consider a linear bandit instance with action set $\mathcal{A} \subseteq \mathbb{R}^d$ and horizon $T$. There exists a universal constant $C$ and a choice for the batch lengths such that Algorithm 1 finishes in at most $M = \lceil \log \log T \rceil + 1$ batches with regret bounded as*

$$R_T \leq C\gamma\sqrt{T} \log \log T \text{ with probability at least } 1 - \delta, \tag{45}$$

*where $\gamma = 8d\sqrt{C_L(\log(1/\delta) + \log T)}$, $C_L = e^8 d$ and $\delta$ is a parameter. Moreover, if for any $a \in \mathcal{A}$ with $\Delta_a > 0$ we have $\Delta_a \geq \Delta_{\min}$, then there exists a choice of batch lengths so that Algorithm 1 finishes in at most $M = \log_4 T$ batches with regret bounded as*

$$R_T \leq C\frac{\gamma^2}{\Delta_{\min}} \log T \text{ with probability at least } 1 - \delta. \tag{46}$$

*Proof.* Note that in Algorithm 1, we end batch $m$ if the total number of pulls reaches $T_m$. Hence, it is not guaranteed that the number of pulls for arm $a_i$ in batch $m$ reaches $n_m(i)$, which complicates the analysis of the concentration for the least squares estimate parameters. To handle this, we first analyze a variant of Algorithm 1 that completes all $n_m(i)$ pulls for each action $a_i, i \in [d]$. We bound the regret of the variant algorithm when a good event $\tilde{G}$ (that we define later) holds, and show that $\mathbb{P}[\tilde{G}] \geq 1 - \delta$. Then, we show that conditioned on $\tilde{G}$, it holds that $\sum_{i=1}^d n_m(i) \leq T_m$, for all batches $m \in [M]$ (see (77)), which implies that Algorithm 1 coincides with the variant algorithm on $\tilde{G}$ in this case. In the following, we refer to the variant algorithm as Algorithm 1 for simplicity.

To invoke Lemma 1, and hence, Lemma 2, we first verify that the conditions of Lemma 1 hold for all batches $m$. We note that as a result of using the definition of $a_m^\star = \mathcal{O}_{1/T}^+(\mathcal{A}; \theta_m)$, due to the use of an approximate oracle and doing the maximization only over $\mathcal{A}$ (not the bigger set $\mathcal{A}'$), the value of $\Delta_m(a)$ can be negative, however, by definition of $\Delta_m = \langle a_m^\star - a_i, \theta_m \rangle$ and the fact that $\mathcal{A}' = \mathcal{A} \cup \mathcal{B}_{1/T}$, we have that

$$\Delta_m(a) \geq -1/T, \quad \forall a \in \mathcal{A}'. \tag{47}$$

Hence, we have that

$$1/2 \overset{(i)}{\leq} 1 - \eta_m/T \overset{(ii)}{\leq} 1 + \eta_m \Delta_m(a)$$
$$\overset{(iii)}{\leq} 1 + 2\eta_m T \overset{(iv)}{\leq} T^2 \tag{48}$$

where $(i)$ follows from $\eta_m = \sqrt{T_{m-1}}/(8\gamma) \leq \sqrt{T}/(8\gamma)$, $(ii)$ follows from (47), $(iii)$ follows from $|\theta_m| = |\mathbf{V}_{m-1}^{-1} \sum_{t=1}^{T_{m-1}} \tilde{a}_t r_t| \leq \sum_{t=1}^{T_{m-1}} \|\tilde{a}_t r_t\|_2 \leq T$ since $\mathbf{V}_m \geq \mathbf{I}, |r_t| \leq 1, \|\tilde{a}_t\|_2 \leq 1$ (recall that $\mathbf{V}_m = \mathbf{I} + \sum_{i=0}^d n_{m-1}(i) a_i a_i^\top \mathbf{1}[a_i \notin \mathcal{B}_{1/T}]$, $\tilde{a}_t$ is the pulled action at the $t$-th iteration of the

previous batch, $n_{m-1}(i)$ is the number of pulls for action $a_i$ in the previous batch, $\{a_i\}_{i=1}^d$ is the set of actions for the approximate optimal design from previous batch), and $(iv)$ uses $\eta_m \leq \sqrt{T}/(8\gamma)$. This shows that Lemma 1 applies to all calls to Algorithm 3, hence, Lemma 2; namely in each batch $m \geq 2$, Algorithm 2 finds an $\exp(4)$ ($C = \exp(1), \alpha = \exp(-3)$) barycentric spanner of the set $\{\phi_m(a)|a \in \mathcal{A}'\}$. For the first batch, we note that Algorithm 1 and Algorithm 2 do not use the same action gaps. Algorithm 2 uses $\theta_1 = 0$ and thus uses $\Delta(a) = 0$ and $\phi(a) = a$. Hence, it finds $\mathcal{C}_1$, an $\exp(4)$-barycentric spanner of $\mathcal{A}'$. Algorithm 1 sets $\Delta_1(a) = 1$, $\forall a \in \mathcal{A}'$, and thus $\tilde{\mathcal{A}}_1 = \{ca|a \in \mathcal{A}'\}$, $c = 1/(1+1/(8\gamma))$ is a scaled version of $\mathcal{A}'$. Hence, $\{\phi_1(a) = ca|a \in \mathcal{C}_1\}$ is a barycentric spanner for $\tilde{\mathcal{A}}_1$ as well. Thus we conclude that for $m = 1$ as well as all other $m \in [M]$, $\{\phi_m(a)|a \in \mathcal{C}_m\}$ is a barycentric spanner for $\tilde{\mathcal{A}}_m$.

We next prove the following lemma that shows the concentration of the estimates $\langle \phi_m(a), \theta_{m+1} \rangle$, $\forall a \in \mathcal{A}'$.

**Lemma 4.** *Let $T \geq 2$, and $\theta_{m+1}$ be the regularized least squares estimate of $\theta_\star$ at the end of batch $m$ in Algorithm 1. Let the event $\mathcal{G}$ be the event*

$$\mathcal{G} : |\langle \phi_m(a), \theta_{m+1} - \theta_\star \rangle| \leq \gamma/\sqrt{T_m}, \quad \forall a \in \mathcal{A}', m \in [M], \tag{49}$$

*where $\gamma = 8d\sqrt{C_L(\log(1/\delta) + \log T)}$. Then, we have that $\mathbb{P}[\mathcal{G}] \geq 1 - \delta$.*

*Proof.* We note that the regularized least squares matrix $\mathbf{V}_{m+1}$ at the end of batch $m$ can be bounded as (recall that the considered variant of Algorithm 1 finishes all $n_m(i)$ pulls $\forall i \in [d]$ and $\forall m \in [M]$)

$$\mathbf{V}_{m+1} \geq \lambda\mathbf{I} + \sum_{i=1}^d \lceil \frac{\pi(i)T_m/8}{(1 + \sqrt{T_{m-1}}\Delta_m(a_i)/(8\gamma))^2} \rceil a_i a_i^\top \mathbf{1}[a_i \notin \mathcal{B}_{1/T}]$$

$$\geq \lambda\mathbf{I} + \sum_{i=1}^d \frac{\pi(i)T_m/8}{(1 + \sqrt{T_{m-1}}\Delta_m(a_i)/(8\gamma))^2} a_i a_i^\top \mathbf{1}[a_i \notin \mathcal{B}_{1/T}]$$

$$= \lambda\mathbf{I} + \sum_{i=1}^d \pi(i)\frac{T_m}{8}\phi_m(a_i)\phi_m(a_i)^\top \mathbf{1}[a_i \notin \mathcal{B}_{1/T}]$$

$$= \lambda\mathbf{I} + \sum_{i=1}^d \pi(i)\frac{T_m}{8}\phi_m(a_i)\phi_m(a_i)^\top - \mathbf{E}, \tag{50}$$

where $\mathbf{E} = \sum_{i=1}^d \pi(i)T_m\phi_m(a_i)\phi_m(a_i)^\top \mathbf{1}[a_i \in \mathcal{B}_{1/T}]$. Hence, using (47), for any $a \in \mathcal{B}_{1/T}, T \geq 2$ we have that

$$\|\phi_m(a)\|_2 = \frac{\|a\|_2}{1 + \Delta_m(a)} \leq \frac{1/T}{1 - 1/T} \leq 2/T. \tag{51}$$

As a result we have that for any $T \geq 2, a \in \mathbb{R}^d$ with $\|a\|_2 \leq 1$

$$a^\top \mathbf{E}a = \sum_{i=1}^d \pi(i)\frac{T_m}{8}a^\top\phi_m(a_i)\phi_m(a_i)^\top a\mathbf{1}[a_i \in \mathcal{B}_{1/T}] \leq \sum_{i=1}^d \pi(i)\frac{T_m}{8}\|a\|_2^2\|\phi_m(a_i)\|_2^2\mathbf{1}[a_i \in \mathcal{B}_{1/T}]$$

$$\leq \sum_{i=1}^d \pi(i)/T \leq 1/T. \tag{52}$$

From (52) and (50) we get that for $T \geq 2$

$$\mathbf{V}_{m+1} \geq \lambda\mathbf{I} + \sum_{i=1}^d \pi(i)\frac{T_m}{8}\phi_m(a_i)\phi_m(a_i)^\top - \mathbf{E} \geq \lambda\mathbf{I} + \sum_{i=1}^d \pi(i)\frac{T_m}{8}\phi_m(a_i)\phi_m(a_i)^\top - 1/T\mathbf{I}$$

$$\overset{(i)}{=} (1 - 1/T)\mathbf{I} + \frac{T_m}{8}\mathbf{V}_{\pi,m} \geq \frac{T_m}{8}\mathbf{V}_{\pi,m}, \tag{53}$$

where $(i)$ follows from $\lambda = 1$ and

$$\mathbf{V}_{\pi,m} = \sum_{i=1}^d \pi(i)\phi_m(a_i)\phi_m(a_i)^\top. \tag{54}$$

By Cauchy-Schwartz inequality we have that

$$|\langle \phi_m(a), \theta_{m+1} - \theta_\star \rangle| \leq \|\phi_m(a)\|_{\mathbf{V}_{m+1}^{-1}} \|\theta_{m+1} - \theta_\star\|_{\mathbf{V}_{m+1}} \overset{(i)}{\leq} \frac{\|\phi_m(a)\|_{\mathbf{V}_{\pi,m}^{-1}}}{\sqrt{T_m/8}} \|\theta_{m+1} - \theta_\star\|_{\mathbf{V}_{m+1}}$$

$$\overset{(ii)}{\leq} 2\sqrt{2C_L d/T_m} \|\theta_{m+1} - \theta_\star\|_{\mathbf{V}_{m+1}}, \tag{55}$$

where $(i)$ follows from (53), and $(ii)$ follows from the fact that $\{\phi_m(a_i), \pi(i)\}_{i=1}^d$ is a $C_L$-approximate design for $\tilde{\mathcal{A}}$. By Theorem 20.5 in [24], we have that with probability at least $1 - \delta$ it holds that

$$\|\theta_{m+1} - \theta_\star\|_{\mathbf{V}_{m+1}} \leq 2\sqrt{\log(1/\delta) + d\log T}, \quad \forall m \in [M]. \tag{56}$$

Combining with (55) we get that the next inequality holds with probability at least $1 - \delta$

$$|\langle \phi_m(a), \theta_{m+1} - \theta_\star \rangle| \leq 4d\sqrt{2C_L(\log(1/\delta) + \log T)/T_m}, \quad \forall m \in [M]. \tag{57}$$

∎

Corollary 1 follows from Lemma 4 and the fact that $1 + \eta_m \Delta_m(a) \geq 1 - 1/\sqrt{T} > 0$ for $T > 1$.

**Corollary 1.** Let $T \geq 2$, and $\theta_{m+1}$ be the regularized least squares estimate of $\theta_\star$ at the end of batch $m$ in Algorithm 1. The following event holds with probability at least $1 - \delta$

$$\mathcal{G}' : |\langle a, \theta_{m+1} \rangle - \mu_a| \leq \frac{\gamma}{\sqrt{T_m}} + \frac{\Delta_m(a)}{8}\sqrt{\frac{T_{m-1}}{T_m}}, \quad \forall a \in \mathcal{A}', m \in [M], \tag{58}$$

where $\gamma = 8d\sqrt{C_L(\log(1/\delta) + \log T)}$ and $\mu_a = \langle a, \theta_* \rangle$.

We introduce the definition of the gap $\Delta_a$ on the set $\mathcal{A}'$ as follows

$$\Delta_a = \sup_{b \in \mathcal{A}} \langle b, \theta_\star \rangle - \langle a, \theta_\star \rangle, \quad \forall a \in \mathcal{A}'. \tag{59}$$

We note that with this definition $\Delta_a$ may be negative for some $a \in \mathcal{A}'$ as the supremum is taken over the smaller set $\mathcal{A}$. However, we have that $\forall a \in \mathcal{A}'$

$$\Delta_a \geq \min\{0, \sup_{b \in \mathcal{A}} \langle b, \theta_\star \rangle - \sup_{u \in \mathcal{B}_{1/T}} \langle u, \theta_\star \rangle\} \geq -1/T. \tag{60}$$

We also have that

$$\Delta_a \leq \max\{1, \sup_{b \in \mathcal{A}} \langle b, \theta_\star \rangle - \inf_{u \in \mathcal{B}_{1/T}} \langle u, \theta_\star \rangle\} \leq 1 + 1/T. \tag{61}$$

We can now prove the following lemma about the concentration of $\Delta_m(a)$.

**Lemma 5.** *Suppose that $\mathcal{G}'$ holds and assume $T_m \geq T_{m-1}$, $\forall m \in [M]$, then we have that the following events hold*

$$\tilde{\mathcal{G}}_m : -4\frac{\gamma}{\sqrt{T_{m-1}}} + \frac{1}{2}\Delta_a \leq \Delta_m(a) \leq 2\Delta_a + 4\frac{\gamma}{\sqrt{T_{m-1}}}, \quad \forall a \in \mathcal{A}', \quad \forall m \in M. \tag{62}$$

*Proof.* We prove the statement by induction on $m$. For $m = 1$ we have that for any $a \in \mathcal{A}'$

$$-4\frac{\gamma}{\sqrt{T_{m-1}}} + \frac{1}{2}\Delta_a \overset{(i)}{=} -4\gamma + \frac{1}{2}\Delta_a \leq \frac{1}{2}\Delta_a \overset{(ii)}{\leq} \frac{1}{2}(1 + 1/T)$$

$$\overset{(iii)}{\leq} \Delta_1(a)$$

$$\overset{(iv)}{\leq} 4\gamma - 2/T \overset{(v)}{\leq} 4\gamma + 2\Delta_a = 4\frac{\gamma}{\sqrt{T_{m-1}}} + 2\Delta_a \tag{63}$$

where $(i)$ uses $T_0 = 1$, $(ii)$ follows from (61), $(iii)$ follows from $\Delta_1(a) = 1$, $(iv)$ uses $\gamma \geq 1$, and $(v)$ follows from (60). Now suppose that $\tilde{\mathcal{G}}_m$ holds. We need to show that $\tilde{\mathcal{G}}_{m+1}$ holds. We have that for any $a \in \mathcal{A}'$

$$\Delta_{m+1}(a) = \langle a_{m+1}^\star - a, \theta_{m+1}\rangle$$
$$\stackrel{(i)}{\leq} \mu_{a_{m+1}^\star} - \mu_a + 2\frac{\gamma}{\sqrt{T_m}} + \left(\frac{\Delta_m(a_{m+1}^\star)}{8} + \frac{\Delta_m(a)}{8}\right)\sqrt{\frac{T_{m-1}}{T_m}}$$
$$= \Delta_a - \Delta_{a_{m+1}^\star} + 2\frac{\gamma}{\sqrt{T_m}} + \left(\frac{\Delta_m(a_{m+1}^\star)}{8} + \frac{\Delta_m(a)}{8}\right)\sqrt{\frac{T_{m-1}}{T_m}}$$
$$\stackrel{(ii)}{\leq} \Delta_a - \Delta_{a_{m+1}^\star} + 2\frac{\gamma}{\sqrt{T_m}} + \left(\frac{2\Delta_{a_{m+1}^\star} + 4\frac{\gamma}{\sqrt{T_{m-1}}}}{8} + \frac{2\Delta_a + 4\frac{\gamma}{\sqrt{T_{m-1}}}}{8}\right)\sqrt{\frac{T_{m-1}}{T_m}}$$
$$= \Delta_a - \Delta_{a_{m+1}^\star} + 3\frac{\gamma}{\sqrt{T_m}} + \left(\frac{\Delta_{a_{m+1}^\star}}{4} + \frac{\Delta_a}{4}\right)\sqrt{\frac{T_{m-1}}{T_m}}$$
$$= 2\Delta_a + 3\frac{\gamma}{\sqrt{T_m}} + \Delta_a\left(1/4\sqrt{\frac{T_{m-1}}{T_m}} - 1\right) + \Delta_{a_{m+1}^\star}\left(1/4\sqrt{\frac{T_{m-1}}{T_m}} - 1\right), \tag{64}$$

where $(i)$ follows from $\mathcal{G}'$, and $(ii)$ follows by the induction hypothesis. We have that if $\Delta_a \geq 0$, then

$$\Delta_a\left(1/4\sqrt{\frac{T_{m-1}}{T_m}} - 1\right) \stackrel{(i)}{\leq} \Delta_a(1/4 - 1) \leq 0, \tag{65}$$

where $(i)$ uses the fact that $T_m \geq T_{m-1}$. If $\Delta_a < 0$, then

$$\Delta_a\left(1/4\sqrt{\frac{T_{m-1}}{T_m}} - 1\right) \leq -\Delta_a \stackrel{(i)}{\leq} 1/T, \tag{66}$$

where $(i)$ follows from (60). Hence, from (65) and (66) we get that

$$\Delta_a\left(1/4\sqrt{\frac{T_{m-1}}{T_m}} - 1\right) \leq 1/T. \tag{67}$$

Similarly, we have

$$\Delta_{a_{m+1}^\star}\left(1/4\sqrt{\frac{T_{m-1}}{T_m}} - 1\right) \leq 1/T. \tag{68}$$

Substituting from (67) and (68) in (64) we get that

$$\Delta_{m+1}(a) \leq 2\Delta_a + 3\frac{\gamma}{\sqrt{T_m}} + 2/T \leq 2\Delta_a + 4\frac{\gamma}{\sqrt{T_m}}, \tag{69}$$

where the last inequality uses $T_m \leq T$ and $\gamma \geq 2$. We next prove a lower bound on $\Delta_{m+1}(a)$. In the following we assume that $\sup_{a \in \mathcal{A}} \mu_a$ is attained by $a^\star \in \mathcal{A}$, and $\sup_{a \in \mathcal{A}'} \langle a, \theta_{m+1}\rangle$ is attained by $\tilde{a}_{m+1}^\star \in \mathcal{A}'$. The proof can be easily extended when the supremums are not attained by using a small approximation and taking the limit. We have that for any $a \in \mathcal{A}'$

$$\Delta_{m+1}(a) = \langle a_{m+1}^\star - a, \theta_{m+1}\rangle \geq \langle \tilde{a}_{m+1}^\star - a, \theta_{m+1}\rangle - 1/T \geq \langle a^\star - a, \theta_{m+1}\rangle - 1/T$$
$$\stackrel{(i)}{\geq} \mu_{a^\star} - \mu_a - 2\frac{\gamma}{\sqrt{T_m}} - \left(\frac{\Delta_m(a^\star)}{8} + \frac{\Delta_m(a)}{8}\right)\sqrt{\frac{T_{m-1}}{T_m}} - 1/T$$
$$= \Delta_a - 2\frac{\gamma}{\sqrt{T_m}} - \left(\frac{\Delta_m(a^\star)}{8} + \frac{\Delta_m(a)}{8}\right)\sqrt{\frac{T_{m-1}}{T_m}} - 1/T$$
$$\stackrel{(ii)}{\geq} \Delta_a - 2\frac{\gamma}{\sqrt{T_m}} - \left(\frac{2\Delta_{a^\star} + 4\frac{\gamma}{\sqrt{T_{m-1}}}}{8} + \frac{2\Delta_a + 4\frac{\gamma}{\sqrt{T_{m-1}}}}{8}\right)\sqrt{\frac{T_{m-1}}{T_m}} - 1/T$$
$$= \Delta_a - 3\frac{\gamma}{\sqrt{T_m}} - \frac{\Delta_a}{4}\sqrt{\frac{T_{m-1}}{T_m}} - 1/T = \frac{1}{2}\Delta_a - 3\frac{\gamma}{\sqrt{T_m}} + \Delta_a\left(\frac{1}{2} - \frac{1}{4}\sqrt{\frac{T_{m-1}}{T_m}}\right) - 1/T. \tag{70}$$

where $(i)$ follows from $\mathcal{G}'$, and $(ii)$ follows by the induction hypothesis. We have that if $\Delta_a \geq 0$, then

$$\Delta_a \left( \frac{1}{2} - \frac{1}{4} \sqrt{\frac{T_{m-1}}{T_m}} \right) \overset{(i)}{\geq} \frac{1}{4} \Delta_a \geq 0, \tag{71}$$

where $(i)$ follows from $T_m \geq T_{m-1}$. If $\Delta_a \leq 0$, then

$$\Delta_a \left( \frac{1}{2} - \frac{1}{4} \sqrt{\frac{T_{m-1}}{T_m}} \right) \geq \frac{1}{2} \Delta_a \geq -\frac{1}{2} 1/T. \tag{72}$$

Substituting from (71) and (72) in (70) we get that

$$\Delta_{m+1}(a) \geq \frac{1}{2} \Delta_a - 3 \frac{\gamma}{\sqrt{T_m}} - 2/T \geq \frac{1}{2} \Delta_a - 4 \frac{\gamma}{\sqrt{T_m}}, \tag{73}$$

where the last inequality uses $T_m \leq T$ and $\gamma \geq 2$. Combining (69) and (73) we get that $\tilde{\mathcal{G}}_{m+1}$ holds. We conclude by induction that $\tilde{\mathcal{G}}_m$ holds for all $m \in [M]$. ∎

We are now ready to prove the regret bound. We first upper bound the regret in batch $m$

$$R^{(m)} = \sum_{t \in H_m} \sup_{a \in \mathcal{A}} \mu_a - \mu_{a_t}, \tag{74}$$

where $H_m$ is the set of time slots for batch $m$, and $a_t$ is the action pulled at time $t$. The following lemma gives a bound on $R^{(m)}$.

**Lemma 6.** *Suppose that $\tilde{\mathcal{G}}_m$ holds, then we have that*

$$R^{(m)} \leq d + 1 + \frac{68\gamma T_m}{\sqrt{T_{m-1}}}. \tag{75}$$

*Moreover, if $\forall a \in \mathcal{A}$ with $\Delta_a > 0$ we have $\Delta_a \geq \Delta_{\min}$ then*

$$R^{(m)} \leq d + 1 + \frac{544\gamma^2 T_m}{\Delta_{\min} T_{m-1}}. \tag{76}$$

*If $T_1 \geq 2d$ then*

$$\sum_{i=1}^{d} n_m(i) \leq T_m. \tag{77}$$

*Proof.* Let $\{\phi_m(a_i), \pi(i)\}_{i=1}^{d}$ be the $C_L$-approximate design at batch $m$ and $a_0 = a_m^\star$. The regret at batch $m$ can be bounded as

$$R^{(m)} \leq T_m \Delta_{a_0} + \sum_{i=1}^{d} n_m(a_i) \Delta_{a_i} \mathbf{1}[a_i \notin \mathcal{B}_{1/T}] \tag{78}$$

We first modify the first term in (78) to put it in the same form of the terms inside the summation. Towards that, we expand the definition of $n_m(i)$ to include $a_0$ by letting $\pi(0) = 16$ ($n_m(0)$ and $\pi(0)$ are values used only for analysis and may not reflect the actual number of pulls for action $a_0$) and

$$n_m(0) = \left\lceil \frac{\pi(0) T_m/8}{(1 + \sqrt{T_{m-1}} \Delta_m(a_0)/(8\gamma))^2} \right\rceil. \tag{79}$$

By definition of $a_0 = a_m^\star$ we also have that $\Delta_m(a_0) \leq 1/T$. Hence, we have that

$$\frac{1}{(1 + \sqrt{T_{m-1}} \Delta_m(a_0)/(8\gamma))^2} \geq 1/2. \tag{80}$$

Substituting in (78), and using $\pi(0) = 16$, we get that

$$R^{(m)} \leq \sum_{i=0}^{d} n_m(a_i) \Delta_{a_i} \mathbf{1}[a_i \notin \mathcal{B}_{1/T}] \tag{81}$$

We notice that on $\tilde{G}_m$ we have

$$n_m(i) = \lceil \frac{\pi(i)T_m/8}{(1+\sqrt{T_{m-1}}\Delta_m(a_i)/(8\gamma))^2} \rceil \leq 1 + \frac{\pi(i)T_m/8}{(1+\sqrt{T_{m-1}}\Delta_m(a_i)/(8\gamma))^2}$$

$$\leq 1 + \frac{\pi(i)T_m/8}{(1+\sqrt{T_{m-1}}(1/2\Delta_{a_i} - 4\gamma/\sqrt{T_{m-1}})/(8\gamma))^2}$$

$$= 1 + \frac{\pi(i)T_m/8}{(1/2 + 1/16\sqrt{T_{m-1}}\Delta_{a_i}/\gamma)^2} \tag{82}$$

This implies that

$$n_m(i) \leq 1 + \min\{T_m/2, \frac{32\gamma^2 T_m}{T_{m-1}\Delta_{a_i}^2}\}\pi(i) \tag{83}$$

The last part of the lemma follows from (83) since $\sum_{i=1}^d n_m(i) \leq d + T_m/2 \sum_{i=1}^d \pi(i) = d + T_m/2 \leq T_m$, where the last inequality follows from $T_1 \geq 2d$. Substituting in (81), we get that

$$R^{(m)} \leq \sum_{i=0}^d n_m(a_i)\Delta_{a_i}\mathbf{1}[a_i \notin \mathcal{B}_{1/T}]$$

$$\leq d + 1 + \sum_{i=0}^d \pi(i)\min\{\Delta_{a_i}T_m/2, \frac{32\gamma^2 T_m}{\Delta_{a_i}T_{m-1}}\}\mathbf{1}[a_i \notin \mathcal{B}_{1/T}] \tag{84}$$

Hence, we have that

$$R^{(m)} \leq d + 1 + \sum_{i=0}^d \pi(i) \sup_{\Delta_{a_i}\geq 0}\min\{\Delta_{a_i}T_m/2, \frac{32\gamma^2 T_m}{\Delta_{a_i}T_{m-1}}\}\mathbf{1}[a_i \notin \mathcal{B}_{1/T}]$$

$$\leq d + 1 + \sup_{\Delta\geq 0}\min\{\Delta T_m/2, \frac{32\gamma^2 T_m}{\Delta T_{m-1}}\}\sum_{i=0}^d \pi(i)$$

$$= d + 1 + 17\sup_{\Delta\geq 0}\min\{\Delta T_m/2, \frac{32\gamma^2 T_m}{\Delta T_{m-1}}\}. \tag{85}$$

We have that $\min\{\Delta T_m/2, \frac{32\gamma^2 T_m}{\Delta T_{m-1}}\}$ is maximized when $\Delta T_m/2 = \frac{32\gamma^2 T_m}{\Delta T_{m-1}}$, hence, when $\Delta = \frac{8\gamma}{\sqrt{T_{m-1}}}$. Substituting in (85) we get that

$$R^{(m)} \leq d + 1 + \frac{68\gamma T_m}{\sqrt{T_{m-1}}}. \tag{86}$$

To prove the gap dependent bound on $R^{(m)}$ we start from (84). We have that if $\Delta_a \geq \Delta_{\min}\forall a \in \mathcal{A} : \Delta_a > 0$, then

$$R^{(m)} \leq d + 1 + \sum_{i=0}^d \pi(i)\min\{\Delta_{a_i}T_m/2, \frac{32\gamma^2 T_m}{\Delta_{a_i}T_{m-1}}\}\mathbf{1}[a_i \notin \mathcal{B}_{1/T}]$$

$$\leq d + 1 + \frac{32\gamma^2 T_m}{\Delta_{\min}T_{m-1}}\sum_{i=0}^d \pi(i)$$

$$\leq d + 1 + \frac{544\gamma^2 T_m}{\Delta_{\min}T_{m-1}} \tag{87}$$

This concludes the proof of the lemma. ∎

To combine the regret across different batches we notice that since $\sum_{m=1}^M T_m \geq T$, Algorithm 1 will finish in at most $M$ batches. The following result follows from Lemma 6.

**Lemma 7.** *Suppose $T_m \geq T_{m-1}, \forall m \in [M]$, $\sum_{m=1}^{M} T_m \geq T$ and $T_0 = 1, T_1 \geq 2d$, then there exists a universal constant $C$ such that with probability at least $1 - \delta$ the regret of Algorithm 1 is bounded as*

$$R_T \leq C \sum_{m=1}^{M} \frac{\gamma T_m}{\sqrt{T_{m-1}}}, \tag{88}$$

*where $\gamma = 8d\sqrt{C_L(\log(1/\delta) + \log T)}$. Moreover, if $\forall a \in \mathcal{A}$ with $\Delta_a > 0$ we have $\Delta_a \geq \Delta_{\min}$ then with probability at least $1 - \delta$ the regret of Algorithm 1 is bounded as*

$$R_T \leq C \sum_{m=1}^{M} \frac{\gamma^2 T_m}{\Delta_{\min} T_{m-1}}. \tag{89}$$

Finally, we use the two sets of batch lengths proposed in [13]. The first set of batch lengths is suitable for worst case regret bounds. We choose the following batch lengths $\{T_m\}$:

$$T_m = \max\{\lfloor T^{1-2^{-m}} \rfloor, 2d\}, m \in [M-1], T_M = T, M = \lceil \log\log T \rceil + 1. \tag{90}$$

We note that $\sum_{m=1}^{M} T_m \geq T$, however, Algorithm 1 finishes whenever the number of rounds reaches $T$, hence, the number of batches is upper bounded by $M$. We also notice that $T_1 \geq 2d, T_m \geq T_{m-1} \forall m \in [M]$. To prove the first regret bound we observe that for $T \geq 2$ and $2 \leq m \leq M - 1$ we have

$$\frac{T_m}{\sqrt{T_{m-1}}} \leq \frac{\lfloor T^{1-2^{-m}} \rfloor}{\sqrt{\lfloor T^{1-2^{-m+1}} \rfloor}} \leq \frac{T^{1-2^{-m}}}{\sqrt{\lfloor T^{1-2^{-m+1}} \rfloor}} = \frac{\sqrt{T}\sqrt{T^{1-2^{-m+1}}}}{\sqrt{\lfloor T^{1-2^{-m+1}} \rfloor}} \leq 2\sqrt{T}. \tag{91}$$

We also have that

$$\frac{T_M}{\sqrt{T_{M-1}}} = \frac{T}{\lfloor T^{1-2^{-\log\log T}} \rfloor} = \frac{T}{\lfloor T/2 \rfloor} \leq 4. \tag{92}$$

Hence, in all cases we have $\frac{T_m}{\sqrt{T_{m-1}}} \leq 4\sqrt{T}$. The regret bound follows by noticing that the regret of the first batch can be bounded by $T_1 \leq \max\{2d, \sqrt{T} + 1\}$ and substituting in (88).

The second set of batch lengths $\{T_m\}$ is suitable for gap dependent regret bounds. We choose the following batch lengths

$$T_m = d4^m, m \in [M], M = \lceil \log_4 T \rceil. \tag{93}$$

We notice $T_1 \geq 2d, T_m \geq T_{m-1} \forall m \in [M], \sum_{m=1}^{M} T_m \geq T$ (Algorithm 1 finishes whenever the number of rounds reaches $T$, hence, the number of batches is upper bounded by $M$). The gap dependent regret bound directly follows by substituting the batch lengths from (93) in (89). ∎

## E Proof of Theorem 2: complexity of Algorithm 1

**Theorem.** *Algorithm 1 finishes in $\tilde{O}(Td^2 + d^4M + \mathcal{T}_{opt}d^3M)$ runtime and uses $\tilde{O}(d^2 + \mathcal{M}_{opt})$ memory, where $\mathcal{T}_{opt}, \mathcal{M}_{opt}$ are the time and space complexity of the linear optimzation oracle for the action set $\mathcal{A}$.*

*Proof.* We notice that the runtime and space complexity of LW-ArgMax is

$$\mathcal{T}_{\text{LW-ArgMax}} = O((d + \mathcal{T}_{opt})\log^3 T), \mathcal{M}_{\text{LW-ArgMax}} = O(d\log^3 T + \mathcal{M}_{opt}). \tag{94}$$

We next upper bound the complexity of Algorithm 2. As the matrix $\mathbf{A}$ is invertible in all iterations, we can use the rank-one update formula of the determinent [29] to perform steps 5 and 10 in $O(d)$ runtime and $O(d^2)$ space complexity. Namely

$$\begin{aligned}
\det(a, \mathbf{A}_{-i}) = \det(\mathbf{A} + (a - a_i)e_i^\top) &= \det(A)(1 + e_i^\top \mathbf{A}^{-1}(a - a_i)) \\
&= \langle a, \det(\mathbf{A})(\mathbf{A}^{-1})^\top e_i \rangle + \det(\mathbf{A})(1 - e_i^\top \mathbf{A}^{-1}a_i) \\
&= \langle a, \det(\mathbf{A})(\mathbf{A}^{-1})^\top e_i \rangle,
\end{aligned} \tag{95}$$

where the last step follows by noticing that the formula is valid for $a = 0$, $\tilde{a}_i$ is the $i$-th column of $\mathbf{A}$. This requires the inverse of matrix $\mathbf{A}$ which can be computed using rank-one updates in $O(d^2)$ time and $O(d^2)$ space [34]

$$(\mathbf{A} + (a - \tilde{a}_i)e_i^\top)^{-1} = \mathbf{A}^{-1} - \frac{\mathbf{A}^{-1}(a - \tilde{a}_i)e_i^\top \mathbf{A}^{-1}}{1 + e_i \mathbf{A}^{-1}(a - \tilde{a}_i)}. \tag{96}$$

We notice that for each repetition of the second for loop, $\mathbf{A}^{-1}$ is updated once while $\det(\mathbf{A})$ can be updated at most $d$ times. Hence, the time and space complexity of one repetition of the for loop in Algorithm 2 is $O(\mathcal{T}_{\text{LW-ArgMax}}d + d^2)$, $O(\mathcal{M}_{\text{LW-ArgMax}} + d^2)$ respectively. By Lemma 2, the for loops is repeated at most $O(d^2 \log d)$ times. Hence, the time and space complexity of Algorithm 2 can be bounded as

$$\mathcal{T}_{\text{LWS}} = O\left((d^4 + \mathcal{T}_{\text{opt}}d^3)\log d \log^3 T\right), \mathcal{M}_{\text{LWS}} = O\left(d^2 \log^3 T + \mathcal{M}_{\text{opt}}\right). \tag{97}$$

We next upper bound the time and space complexity of Algorithm 1.

- The time and space complexity of finding the barycentric spanner in step 5 is $\mathcal{T}_{\text{LWS}}, \mathcal{M}_{\text{LWS}}$ respectively.

- The computation of the least squares matrix requires $O(T_m d^2)$ time and $O(d^2)$ space, while its inversion requires $O(d^3)$ runtime. Hence, $\theta_m$ can be computed in $O(T_m d^2 + d^3)$ time and $O(d^2)$ space.

- Computing the estimated best action in step 11 requires $\mathcal{T}_{\text{opt}}, \mathcal{M}_{\text{opt}}$ time and space respectively.

Hence, in total Algorithm 1 runtime is $O(Td^2 + (d^4 + \mathcal{T}_{\text{opt}}d^3)M \log d \log^3 T)$ while the space complexity is $O\left(d^2 \log^3 T + \mathcal{M}_{\text{opt}}\right)$. ∎

## F   Approximate oracle over $\mathcal{X}_m$

**Lemma 8.** *Consider a given $m \in [M]$ and let $g^{(m)}(\theta) = \frac{1}{|H_{m-1}|}\sum_{t \in H_{m-1}} \mathcal{O}(\mathcal{A}_t; \theta)$, $\mathcal{X}_m = \{g^{(m)}(\theta)|\theta \in \Theta'\}$, where $H_m$ is the set of indices for rounds in batch $m$ and $\Theta' = [\theta]_q|\theta \in \Theta$ is a discretization of $\Theta$, $[\theta]_q = q\lfloor\theta\sqrt{d}/q\rfloor/\sqrt{d}$ and $q$ is the discretization parameter. For any $\theta \in \mathcal{S}_1, \epsilon \in \mathbb{R}^+$, if $q \le \epsilon/2$, we have that*

$$\langle g^{(m)}([\theta]_q), \theta\rangle \ge \sup_{a \in \mathcal{X}_m} \langle a, \theta\rangle - \epsilon. \tag{98}$$

*Proof.* We first observe that

$$0 \le \theta - [\theta]_q = \theta - \frac{\lfloor\theta\sqrt{d}/q\rfloor}{\sqrt{d}/q} \le q/\sqrt{d}\mathbf{1}. \tag{99}$$

It follows that $\|\theta - [\theta]_q\|_2 \le q$. We notice that

$$\langle g^{(m)}(\theta), \theta\rangle = \frac{1}{|H_{m-1}|}\sum_{t \in H_{m-1}}\langle \mathcal{O}(\mathcal{A}_t; \theta), \theta\rangle \ge \frac{1}{|H_{m-1}|}\sum_{t \in H_{m-1}}\langle \mathcal{O}(\mathcal{A}_t; \theta'), \theta\rangle$$

$$= \langle g^{(m)}(\theta'), \theta\rangle \forall \theta' \in \Theta. \tag{100}$$

Hence,

$$\langle g^{(m)}(\theta), \theta\rangle \ge \sup_{\theta' \in \Theta'}\langle g^{(m)}(\theta'), \theta\rangle. \tag{101}$$

We also have that

$$\langle g^{(m)}([\theta]_q), \theta\rangle = \langle g^{(m)}([\theta]_q), [\theta]_q\rangle + \langle g^{(m)}([\theta]_q), \theta - [\theta]_q\rangle$$

$$\ge \langle g^{(m)}([\theta]_q), [\theta]_q\rangle - \|g^{(m)}([\theta]_q)\|_2\|\theta - [\theta]_q\|_2$$

$$\ge \langle g^{(m)}([\theta]_q), [\theta]_q\rangle - q$$

---
**Algorithm 4** Efficient Batched Algorithm for contextual linear bandits
---

1: Input: number of batches $M$, batch lengths $\{T_m\}_{m=1}^M$, horizon $T$, confidence parameter $\delta$, set of unknown parameters $\Theta \subseteq \mathcal{B}_1$, discretization parameter $q$.

2: Select modified batch lengths $\{\tau_m\}_{m=1}^{2M}$ to $\tau_m = T_{m//2}$, where $//$ is the integer division.

3: Let $C_L = \exp(8)d$, $\gamma = 10\sqrt{C_L d(\log(8M/\delta) + 57d\log^2(6T))}$, $\tau_{-1} = \tau_0 = 1$, $\Theta' = \{[\theta]_q = \lfloor \theta/(\sqrt{d}q)\rfloor \sqrt{d}q | \theta \in \Theta\}$.

4: Let $g^{(1)} : \Theta' \to \mathbb{R}^d$ be defined as $g^{(1)}(\theta) = 0$, $\forall \theta \in \Theta'$, and let $\mathcal{X}_1 = \{g^{(1)}(\theta) | \theta \in \Theta'\}$, $\mathcal{X}_1' = \mathcal{X}_1 \cup \mathcal{B}_{1/T}$, $\Delta_1(a) = 1 \ \forall a \in \mathcal{X}_1'$.

5: Initialize: $\theta_1 = 0$, $a_1^\star$ to be a random action in $\mathcal{X}_1$.

6: **for** $m = 1 : 2M$ **do**

7:     Calculate $\{a_i, \theta^{(i)}\}_{i=1}^d = \text{LWS}(\mathcal{X}_m', \eta_m = \sqrt{\tau_{m-2}}/(8\gamma), a_m^\star, \theta_m)$, where $a_i = g^{(m)}(\theta^{(i)})$.[9]

8:     Let $\pi(i) = 1/d \quad \forall i \in [d]$, $a_0 = a_m^\star = g^{(m)}(\theta^{(0)})$, where $\theta^{(0)} = [\theta_m]_q = q\lfloor \theta_m \sqrt{d}/q\rfloor/\sqrt{d}$.

9:     **for** $i = 1 : d$ **do**

10:         If $a_i \notin \mathcal{B}_{1/T}$, calculate $\Delta_m(a_i) = \langle a_m^\star - a_i, \theta_m\rangle$ and play $a = \mathcal{O}(\mathcal{A}_t; \theta^{(i)})$, $n_m(i) = \lceil \frac{\pi(i)\tau_m/4}{(1+\sqrt{\tau_{m-1}}\Delta_m(a_i)/(8\gamma))^2}\rceil$ times. **go to** step 12 if the number of pulls in the current batch reaches $\tau_m$. Terminate Algorithm 1 if the total number of pulls reaches $T$.

11:     play $a = \mathcal{O}(\mathcal{A}_t; \theta^{(0)})$ for $\max\{0, \tau_m - \sum_{i=1}^d n_m(i)\}$ times.

12:     Compute the regularized (with $\lambda = 1$) least squares estimator $\mathbf{V}_m = \mathbf{I} + \sum_{i=1}^{\tau_m} a_i a_i^\top$ and $\theta_{m+1} = \mathbf{V}_m^{-1}\sum_{i=1}^{\tau_m} r_i a_i$.

13:     Update $a_{m+1}^\star = \mathcal{O}_{1/T}^+(\mathcal{X}_m; \theta_{m+1})$.

14:     Let $g^{(m+1)}(\theta) = \frac{1}{\tau_m}\sum_{t \in H_m}\mathcal{O}(\mathcal{A}_t; \theta)$, $\mathcal{X}_{m+1} = \{g^{(m+1)}(\theta) | \theta \in \Theta'\}$, $\mathcal{X}_{m+1}' = \mathcal{X}_{m+1} \cup \mathcal{B}_{1/T}$, where $H_m$ is the set of indices of the rounds in batch $m$.

---

$$\overset{(i)}{\geq} \langle g^{(m)}(\theta), [\theta]_q\rangle - q$$
$$\geq \langle g^{(m)}(\theta), \theta\rangle - \|g^{(m)}(\theta)\|_2\|\theta - [\theta]_q\|_2 - q$$
$$\geq \langle g^{(m)}(\theta), \theta\rangle - 2q \overset{(ii)}{\geq} \sup_{\theta' \in \Theta'}\langle g^{(m)}(\theta'), \theta\rangle - 2q = \sup_{a \in \mathcal{X}_m}\langle a, \theta\rangle - 2q, \quad (102)$$

where $(i)$ and $(ii)$ follow from (101). ∎

## G   Pseudo-code of efficient batched algorithm for contextual linear bandits

The pseudo-code for our algorithm for linear contextual bandits is provided in Algorithm 4. The algorithm follows similar steps to Algorithm 1 with the following exceptions. The set of actions $\mathcal{X}_m$ is updated (see step 14) in every batch using contexts observed in the previous batch. It is important to note that these sets $\mathcal{X}_m$ (in steps 4 and 14) are never actually computed; the definitions are provided for notation purposes. We only need an approximate optimizer for the set $\mathcal{X}_m$ to construct the approximate barycentric spanner in Algorithm 2. As shown in App. F, $g^{(m)}([\theta]_q)$ for sufficiently small $q$ can serve as our approximate oracle. Furthermore, the computation of $g^{(m)}([\theta]_q)$ can be performed using $O(T)$ calls to the linear optimization oracle $\mathcal{O}(\mathcal{A}_t; .)$, hence, with complexity of $O(T\mathcal{T}_{\text{opt}})$.

Additionally, we assume that LW-ArgMax (and LWS) returns for each $a_i$ the value $\theta^{(i)} = [\tilde{\theta}_i]_q$. Here, $\tilde{\theta}_i$ is the input to the approximate linear optimization oracle which yielded the output $a_i$. The final difference from Algorithm 1 is that we do not play action $a_i$ for $n_m(i)$ times (note that $a_i$ may not be in $\mathcal{A}_t$); instead we play policy $\mathcal{O}(\mathcal{A}_t; \theta^{(i)})$ for $n_m(i)$ times, where $\theta^{(i)}$ is the parameter associated with $a_i$ returned by LWS (Algorithm 2) as described earlier.

---

[9] Recall that in the contextual setting we assume that LW-ArgMax (and LWS) returns for each $a_i$ the value $\theta^{(i)} = [\tilde{\theta}_i]_q$, where $\tilde{\theta}_i$ is the input to the approximate linear optimization oracle that resulted in the output $a_i$.

# H    Proof of Theorem 3: regret analysis for contextual bandits

**Theorem 3.** *Consider a contextual linear bandit instance with $\mathcal{A}_t$ generated from an unknown distribution $\mathcal{D}$. There exists a universal constant $C$ and a choice for batch lengths such that Algorithm 4 , with $q = (1 - \exp(-1))/(24T^{7+12\log T})$, finishes in $O(\log \log T)$ batches with regret upper bounded as*

$$R_T \leq C\gamma\sqrt{T}\log\log T$$

*with probability at least $1 - 2\delta$, where $\gamma = 10\sqrt{C_L d(\log(8M/\delta) + 57d\log^2(6T))}$. Moreover, the running time and space complexity are $\tilde{O}(d^4 + \mathcal{T}_{opt}d^3 T)$ and $\tilde{O}(d^2 + \mathcal{M}_{opt})$, respectively.*

*Proof.* Recall that at each round $t$, Algorithm 4 pulls action $a_t$ associated with a value $\theta_t$ (see step 7 in Algorithm 4). To upper bound the regret, we follow a technique proposed in [18] by first upper bounding the quantity

$$R_T^L = \sum_{t=1}^{T} \sup_{\theta \in \Theta} \langle g(\theta) - g(\theta_t), \theta_\star \rangle \tag{103}$$

which can be thought of as the regret of the algorithm on a reduced linear bandit instance [18]. Then we can use Theorem 1 in [18] which states that $|R_T - R_T^L| = \tilde{O}(\sqrt{T})$ with high probability to upper bound the regret $R_T$.

As in the proof of Theorem 1 instead of analyzing Algorithm 4 which ends batch $m$ if the total number of pulls reaches $T_m$, we analyze a variant algorithm that completes all the required pulls of the actions in the barycentric spanner. We bound the regret of the variant algorithm when a good event $\tilde{G}$ (that we define later) holds, and show that $\mathbb{P}[\tilde{G}] \geq 1 - \delta$. Then, we show that conditioned on $\tilde{G}$, it holds that $\sum_{i=1}^{d} n_m(i) \leq \tau_m$, for all batches $m \in [2M]$ (see (115)), which implies that Algorithm 4 coincides with the variant algorithm on $\tilde{G}$ in this case. We also refer to the variant algorithm as Algorithm 4 for simplicity.

Recall that

$$g(\theta) = \mathbb{E}_{\mathcal{A} \sim \mathcal{D}}[\mathcal{O}(\mathcal{A}; \theta)], g^{(m)}(\theta) = \frac{1}{|H_{m-1}|} \sum_{t \in H_{m-1}} \mathcal{O}(\mathcal{A}_t; \theta), \mathcal{X}_m = \{g^{(m)}(\theta)|\theta \in \Theta'\}, \tag{104}$$

where $H_m$ is the set of indices for the rounds in batch $m$ and $\Theta' = \{[\theta]_q|\theta \in \Theta\}$ is a discretization of $\Theta$, $[\theta]_q = q\lfloor\theta\sqrt{d}/q\rfloor/\sqrt{d}$ and $q$ is the discretization parameter. Recall also that $\mathbf{V}_m$ is the regularized least squares matrix in step 12 of Algorithm 4 with $\lambda = 1$, and denote $\epsilon_m = \sup_{\theta' \in \Theta', \theta \in \Theta} |\langle g^m(\theta') - g(\theta'), \theta \rangle|$ in the extended reals $\mathbb{R} \cup \{\infty\}$. We also denote $\epsilon(t) = \langle g(\theta^{(t)}) - g^{(m)}(\theta^{(t)}), \theta_\star \rangle$, where $g^{(m)}(\theta^{(t)}) \in \mathcal{X}_m, r_t$ are the action and reward at iteration $t$ of batch $m$. We first upper bound the error in estimating $\mu_{g(\theta')} = \langle g(\theta'), \theta_\star \rangle$ for an action $g(\theta')$ at the end of batch $m$ for $\theta' \in \Theta'$. We have that for any $a \in \mathbb{R}^d$, $|\langle a, \theta_{m+1} - \theta_\star \rangle|$ can be decomposed as

$$|\langle a, \theta_{m+1} - \theta_\star \rangle| = |\langle a, \mathbf{V}_m^{-1}\sum_{t=1}^{\tau_m} r_t a_t - \theta_\star \rangle| \stackrel{(i)}{=} |\langle a, \mathbf{V}_m^{-1}\sum_{t=1}^{\tau_m}(\theta_\star^\top g(\theta^{(t)}) + \eta_t')g^{(m)}(\theta^{(t)}) - \theta_\star \rangle|$$

$$= |\langle a, \mathbf{V}_m^{-1}\sum_{t=1}^{\tau_m}(\theta_\star^\top g^{(m)}(\theta^{(t)}) + \epsilon(t) + \eta_t')g^{(m)}(\theta^{(t)}) - \theta_\star \rangle|$$

$$= |\langle a, \mathbf{V}_m^{-1}\left((\mathbf{V}_m - \mathbf{I})\theta_\star + \sum_{t=1}^{\tau_m}(\epsilon(t) + \eta_t')g^{(m)}(\theta^{(t)})\right) - \theta_\star \rangle|$$

$$= |\langle a, -\mathbf{V}_m^{-1}\theta_\star + \mathbf{V}_m^{-1}\sum_{t=1}^{\tau_m}(\epsilon(t) + \eta_t')g^{(m)}(\theta^{(t)}) \rangle|$$

$$\leq |\langle a, -\mathbf{V}_m^{-1}\theta_\star \rangle| + |\langle a, \mathbf{V}_m^{-1}\sum_{t=1}^{\tau_m}\epsilon(t)g^{(m)}(\theta^{(t)}) \rangle|$$

$$+ |\langle a, \mathbf{V}_m^{-1} \sum_{t=1}^{\tau_m} \eta_t' g^{(m)}(\theta^{(t)})\rangle|$$

$$\leq \|a\|_{\mathbf{V}_m^{-1}} \|\theta_\star\|_{\mathbf{V}_m^{-1}} + |\langle a, \mathbf{V}_m^{-1} \sum_{t=1}^{\tau_m} \epsilon(t) g^{(m)}(\theta^{(t)})\rangle|$$

$$+ |\langle a, \mathbf{V}_m^{-1} \sum_{t=1}^{\tau_m} \eta_t' g^{(m)}(\theta^{(t)})\rangle|$$

$$\overset{(ii)}{\leq} \|a\|_{\mathbf{V}_m^{-1}} + |\langle a, \mathbf{V}_m^{-1} \sum_{t=1}^{\tau_m} \epsilon(t) g^{(m)}(\theta^{(t)})\rangle| + |\langle a, \mathbf{V}_m^{-1} \sum_{t=1}^{\tau_m} \eta_t' g^{(m)}(\theta^{(t)})\rangle|, \tag{105}$$

where $(i)$ follows from Theorem 1 in [18], $\eta_t'$ is a zero mean noise conditioned on the filtration of history and $\theta^{(t)}$ and $(ii)$ uses $\mathbf{V}_m \geq \mathbf{I}$. We next bound the term $|\langle a, \mathbf{V}_m^{-1} \sum_{t \in H_m} \epsilon(t) g^{(m)}(\theta^{(t)})\rangle|$. We have that

$$|\langle a, \mathbf{V}_m^{-1} \sum_{t=1}^{\tau_m} \epsilon(t) g^{(m)}(\theta^{(t)})\rangle| \leq \sqrt{\tau_m \sum_{t=1}^{\tau_m} \epsilon(t)^2 a^\top \mathbf{V}_m^{-1} g^{(m)}(\theta^{(t)}) g^{(m)}(\theta^{(t)})^\top \mathbf{V}_m^{-1} a}$$

$$\overset{(i)}{\leq} \epsilon_m \sqrt{\tau_m \sum_{t=1}^{\tau_m} a^\top \mathbf{V}_m^{-1} g^{(m)}(\theta^{(t)}) g^{(m)}(\theta^{(t)})^\top \mathbf{V}_m^{-1} a}$$

$$\leq \epsilon_m \sqrt{\tau_m a^\top \mathbf{V}_m^{-1} (\mathbf{V}_m - \mathbf{I}) \mathbf{V}_m^{-1} a}$$

$$\leq \epsilon_m \sqrt{\tau_m (\|a\|_{\mathbf{V}_m^{-1}}^2 - \|a\|_{\mathbf{V}_m^{-2}}^2)} \leq \epsilon_m \sqrt{\tau_m (\|a\|_{\mathbf{V}_m^{-1}}^2 - \|a\|_{\mathbf{V}_m^{-2}}^2)}$$

$$\leq \epsilon_m \sqrt{\tau_m (\|a\|_{\mathbf{V}_m^{-1}}^2 - \|a\|_{\mathbf{V}_m^{-2}}^2)} \leq \epsilon_m \sqrt{\tau_m \|a\|_{\mathbf{V}_m^{-1}}^2}, \tag{106}$$

where $(i)$ follows by the definition of $\epsilon_m = \sup_{\theta' \in \Theta', \theta \in \Theta} |\langle g^m(\theta') - g(\theta'), \theta\rangle|$ and $\epsilon(t) = \langle g(\theta^{(t)}) - g^{(m)}(\theta^{(t)}), \theta_\star\rangle$. We have from Theorem 2 in [18] that the following event holds with probability at least $1 - \delta/(4M)$

$$\mathcal{G}_m^\epsilon : \epsilon_m \leq 2\sqrt{\frac{\log(8M|\Theta'|/\delta)}{\tau_{m-1}}}. \tag{107}$$

We also have that from eq. (20.2) of [24] the following event holds with probability at least $1 - \delta/(4M)$

$$\mathcal{G}_m^\eta : |\langle a, \mathbf{V}_m^{-1} \sum_{t=1}^{\tau_m} \eta_t' g^{(m)}(\theta^{(t)})\rangle| \leq \sqrt{2 \sum_{t=1}^{\tau_m} (a^\top \mathbf{V}_m^{-1} g^{(m)}(\theta^{(t)}))^2 \log(4M|\Theta'|/\delta)}$$

$$\overset{(i)}{\leq} \sqrt{2\|a\|_{\mathbf{V}_m^{-1}}^2 \log(4M|\Theta'|/\delta)} \forall a \in \tilde{\mathcal{X}}_m, \tag{108}$$

where $(i)$ follows by expanding $(a^\top \mathbf{V}_m^{-1} g^{(m)}(\theta^{(t)}))^2$ as in (106). From Lemma 8 we have that for $q = (1 - e^{-1})/(24T^{7+12\log T})$, the function $g^{(m)}([\theta]_q)$ is an approximate linear optimization oracle with additive gap at most $(1 - e^{-1})/(12T^{7+12\log T})$. Hence, using Lemma 1[10] and Lemma 2, Algorithm 2 finds a set $\mathcal{C}_m$ such that $\{\phi_m(a)|a \in \mathcal{C}_m\}$ is an $e^8$ approximate spanner for $\tilde{\mathcal{X}}_m$. By the properties of the $C_L$-approximate design, similar to (53) we have that $\|\phi_m(a)\|_{\mathbf{V}_m^{-1}} \leq \sqrt{C_L d/\tau_m} \forall a \in \mathcal{X}_m'$, where $C_L = e^8 d$. Hence, substituting from (106), (107) and (108) in (105) we get that the following holds on $\mathcal{G}_m^\eta \cap \mathcal{G}_m^\epsilon$

$$|\langle \phi_m(a), \theta_{m+1} - \theta_\star\rangle| \leq \sqrt{C_L d/\tau_m} + 4\sqrt{\frac{C_L d \log(8M|\Theta'|/\delta)}{\tau_{m-1}}} \leq 5\sqrt{\frac{C_L d \log(8M|\Theta'|/\delta)}{\tau_{m-1}}} \forall a \in \mathcal{X}_m'. \tag{109}$$

---

[10]The verification of the conditions stated in Lemma 1 is equivalent to the verification conducted at the beginning of the proof of Theorem 1.

We notice that for $q = (1 - e^{-1})/(24T^{7+12\log T})$, we have that $|\Theta'| \leq 6T^{3d(7+12\log T)}$. Hence, $\log|\Theta'| \leq 57d\log^2(6T)$. Hence, $C_L d\log(8M|\Theta'|/\delta) = C_L d(\log(8M/\delta) + 57d\log^2(6T))$. Substituting in (109) we get that the following holds on $\mathcal{G}_m^\eta \cap \mathcal{G}_m^\epsilon$

$$|\langle\phi_m(a), \theta_{m+1} - \theta_\star\rangle| \leq 5\sqrt{C_L d(\log(8M/\delta) + 57d\log^2(6T))/\tau_{m-1}} \leq \frac{\gamma/2}{\sqrt{\tau_{m-1}}}. \tag{110}$$

By definition of $\phi_m$ it follows that the following holds on $\mathcal{G}_m^\eta \cap \mathcal{G}_m^\epsilon$

$$|\langle a, \theta_{m+1} - \theta_\star\rangle| \leq \frac{\gamma/2}{\sqrt{\tau_{m-1}}} + \frac{\Delta_m(a)}{8}\sqrt{\frac{\tau_{m-2}}{\tau_{m-1}}} \forall a \in \mathcal{X}'_m. \tag{111}$$

Hence, by definition of $\mathcal{G}_m^\epsilon$ in (107) the following holds on $\mathcal{G}_m^\eta \cap \mathcal{G}_m^\epsilon$

$$\begin{aligned}
|\langle a, \theta_{m+1}\rangle - \mu_a| &\leq |\langle a, \theta_{m+1} - \theta_\star\rangle| + |\langle a, \theta_\star\rangle - \mu_a| \\
&\leq \frac{\gamma/2}{\sqrt{\tau_{m-1}}} + \frac{\Delta_m(a)}{8}\sqrt{\frac{\tau_{m-2}}{\tau_{m-1}}} + \epsilon_m \\
&\leq \frac{\gamma}{\sqrt{\tau_{m-1}}} + \frac{\Delta_m(a)}{8}\sqrt{\frac{\tau_{m-2}}{\tau_{m-1}}} \forall a \in \mathcal{X}'_m. 
\end{aligned} \tag{112}$$

We recall that $\mathbb{P}[\mathcal{G}_m^\epsilon] \geq 1 - \delta/(4M)$, $\mathbb{P}[\mathcal{G}_m^\eta] \geq 1 - \delta/(4M)$. Hence, by the union bound we have that

$$\mathbb{P}[\tilde{\mathcal{G}}] \geq 1 - \delta, \tilde{\mathcal{G}} = \cap_{m\in[2M]}(\mathcal{G}_m^\eta \cap \mathcal{G}_m^\epsilon) \tag{113}$$

Then, following the proof of Lemma 5 by replacing every $\tau_m$ with $\tau_{m-1}$ and every $\tau_{m-1}$ with $\tau_{m-2}$ we get that the following event hold on $\tilde{\mathcal{G}}$

$$-4\frac{\gamma}{\sqrt{\tau_{m-2}}} + \frac{1}{2}\Delta_a \leq \Delta_m(a) \leq 2\Delta_a + 4\frac{\gamma}{\sqrt{\tau_{m-2}}} \forall a \in \mathcal{A}' \forall m \in M. \tag{114}$$

Hence, following the same steps as in Lemma 6 we get that there is a universal constant $C$ such that the following holds on $\tilde{\mathcal{G}}$

$$R_T^L \leq C\sum_{m=1}^{2M}\frac{\gamma\tau_m}{\sqrt{\tau_{m-2}}} = C\sum_{m=1}^{2M}\frac{\gamma T_{m//2}}{\sqrt{T_{m//2-1}}}, \sum_{i=1}^d n_m(i) \leq \tau_m, \tag{115}$$

where $R_T^L$ is the regret of the algorithm on the linear bandit instance defined in (103). Using the batch lengths in (90), we get, from (91), that the following holds on $\tilde{\mathcal{G}}$

$$R_T^L \leq 8C\gamma\sqrt{T}M \tag{116}$$

From Theorem 1 in [18] we have that $|R_T^L - R_T| \leq \sqrt{T\log(T/\delta)}$ with probability at least $1 - \delta$. By the union bound and triangle inequality it follows that

$$R_T \leq 16C\gamma\sqrt{T}M \tag{117}$$

with probability at least $1 - 2\delta$.

The complexity result follows from Theorem 2 by observing that computing $g^{(m)}([\theta]_q)$ (our approximate oracle) requires at most $T$ calls to $\mathcal{O}(\mathcal{A}_t; .)$. Hence, the time and space complexity of Algorithm 4 are $O((d^4 + \mathcal{T}_{\text{opt}}d^3 T)M\log d\log^3 T)$ and $O(d^2\log^3 T + \mathcal{M}_{\text{opt}})$, respectively. ∎

# I Numerical comparison of complexity of our scheme

In this appendix we present a small experiment to compare the computational complexity of computing the exploration policy of Algorithm 4 versus the complexity of computing the policy in [40] (complexity of one batch). We do not consider other batched algorithms such as [32] since they are not feasible to implement even for a small number of actions. We used $d = 5$ dimensions and a batch of size 100 iterations. For simplicity we use a fixed action set (unit sphere), however, this knowledge

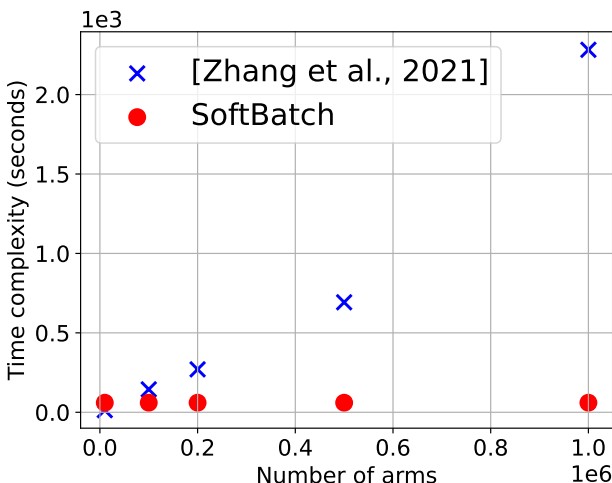

Figure 1: Complexity of computing our exploration policy versus the state of the art complexity.

is not revealed to the algorithms, i.e., the algorithms assume that the action set may change over time. As the policy of [40] requires to solve a non-convex optimization problem, it is not feasible to implement it for infinite number of actions. Instead, we solve the optimization problem over a finite subset of $k$ actions sampled uniformly at random from the action set. In contrast, our algorithm can be directly applied for the infinite action set, hence, the computational complexity will not depend on $k$. In Fig. 1, we plot the time complexity versus the sampled number of actions (on Intel(R) Xeon(R) CPU @ 2.20GHz, 56MB cache). We observe that for moderately large number of actions, our algorithm achieves significant savings in computational complexity as compared to the scheme of [40].