# OpenReview forum: "Efficient Batched Algorithm for Contextual Linear Bandits with Large Action Space via Soft Elimination"
_NeurIPS.cc/2023/Conference — NeurIPS 2023 poster_

### Official Review · Reviewer_6gSX · 2023-06-12

**Soundness:** 3 good
**Presentation:** 2 fair
**Contribution:** 3 good
**Rating:** 6
**Confidence:** 4

**Summary:**

This paper proposes the first efficient batch algorithm for (contextual) linear bandits. Instead of relying on the classical elimination-based algorithmic design, this paper adopts a linear optimization oracle over the large action set. The proposed algorithm achieves $\tilde{O}(\sqrt{}T)$ regret with the number of batches matching the lower bound. The specific algorithm for contextual linear bandit is built upon a reduction technique which reduces contextual linear bandits to bandits. Directly applying this reduction incurs high computational cost. To circumvent this, an approximated linear optimization oracle is designed.

**Strengths:**

1. Meaningful theoretical contribution: the first efficient batch algorithm for contextual linear bandits is proposed and analyzed. The number of batches matches the lower bound.

2. Novel idea in algorithmic design: instead of classical design based on hard elimination, this paper uses an interesting ‘soft’ elimination which eliminates candidates for the unknown model parameters in each batch, incurring low regret.


**Weaknesses:**

I did not detect any major weakness in this paper. However, I do believe this paper would benefit from adding some simple numerical simulation. I understand this is a theoretical paper. But due to the complexity of the theoretical analysis and technical algorithmic design, it would be easier to evaluate the performance of the algorithm via numerical simulations on contextual linear bandits. Specifically, it would be better if it can be demonstrated that the number of batches is indeed of the order $\log \log (T)$. It would also be interesting to compare with some benchmark algorithms for contextual linear bandits not subject to the batch constraint, and see whether the proposed algorithm has inferior regret due to the batch constraint. Simple numerical simulations suffice.
I also have some questions. Please see the Questions section.


**Questions:**

1. I am unclear about what are the major technical novelty/contributions in the reduction discussed in section 4. It seems to me that most ideas follow directly from [18], except for the tuning of the batch length. It would be great if the authors can further elaborate.

2. It is mentioned in line 76 that novel approaches are proposed to bound the regret of the algorithm. Could the authors emphasize/summarize the approaches? The current section 3.2 is quite overwhelming to readers.


**Limitations:**

Yes.

---

> ### Author Rebuttal · Authors · 2023-08-10
>
> We thank the reviewer for their input and for appreciating our contribution.
>
> * **[Numerical simulations]** Indeed, as the reviewer notes, in this work we focused on the theoretical understanding of efficient batch learning in contextual bandits, where we show a batched bandit algorithm that provably achieves computational efficiency while maintaining near-optimal regret bound.
> To illustrate this, let us consider a $d$-dimensional linear bandit problem with $T^d$ actions.
> Our algorithm takes only $\mathrm{poly}(dT)$ time per batch whereas previous batched algorithms can take $\Omega(T^d)$ time, exponential in $d$.\
> \
> Having said that, following the reviewer's suggestion, we have done a small experiment to compare the computational complexity of computing the exploration policy of our algorithm versus the complexity of computing the policy in [Zhang et al., 2021] (complexity of one batch). We do not consider other batched algorithms such as [Ruan et al., 2021] since they are not feasible to implement even for a small number of arms. We used $d=5$ dimensions and a batch of size $100$ iterations. For simplicity we use a fixed action set (unit sphere), however, this knowledge is not revealed to the algorithms, i.e., the algorithms assume that the action set may change over time.
> As the policy of [Zhang et al., 2021] requires to solve a non-convex optimization problem, it is not feasible to implement it for infinite number of arms. Instead, we solve the optimization problem over a finite subset of $k$ arms sampled uniformly at random from the action set. In contrast, our algorithm can be directly applied for the infinite action set, hence, the computational complexity will not depend on $k$. In the figure attached to the common rebuttal, we plot the time complexity versus the sampled number of arms (on Intel(R) Xeon(R) CPU @ 2.20GHz, 56MB cache). We observe that for moderately large number of arms, our algorithm achieves significant savings in computational complexity as compared to the scheme of [Zhang et al., 2021].\
> \
> We note again that theoretically we have proved the same order regret bounds; but we will be happy to add numerical evaluation as well in the next version.
>
> * **[Number of batches]** Regarding simulations to calculate the number of batches, we note that by construction of our algorithm, and in particular the choice of batch lengths in equation (90) in Appendix D, the number of batches  our algorithms use is always bounded by $2\log\log T + 1$.
> * **[Novelty in the reduction]** Thank you for this question. Please refer to the paragraph with title ``Novelty in the reduction" in the common rebuttal for the answer to this question.
>
> * **[Approach to bound the regret]** We provide a summary of the novelty of our algorithm and analysis in the common rebuttal. Please refer to the paragraph with title ``Novelty in bounding the regret". If the reviewer are interested, we highly encourage you to dig more into the details of the algorithm, and we are happy to address any additional questions you may have.

---

> > ### Comment · Reviewer_6gSX · 2023-08-14
> > **Response to the rebuttal**
> >
> > I would like to thank the authors for the rebuttal. I appreciate the description on the numerical experiment. Please add the numerical results to the updated version.

---

> > > ### Author Response · Authors · 2023-08-17
> > >
> > > Thank you for your response and the helpful comments. We will add the numerical results to the updated version. Please let us know if you have any additional questions or concerns.

---

### Official Review · Reviewer_DZuy · 2023-07-06

**Soundness:** 3 good
**Presentation:** 4 excellent
**Contribution:** 3 good
**Rating:** 7
**Confidence:** 3

**Summary:**

This paper proposes an algorithm for the batched (contextual) linear bandit problem with a large action space. The proposed algorithm has low time and space complexity. The proposed algorithm for the batched linear bandit problem selects arms in each batch according to a G-optimal design based on a modified action set and chooses them a number of times, dependent on the estimated sub-optimality gap. As a result, it achieves efficient time and computational complexity, even when the number of actions is large, unlike the existing elimination algorithms. The extension to contextual linear bandits is achieved by modifying part of the existing reduction for linear bandits. The proposed algorithm achieves a near-optimal regret with the optimal number of batches. Although there is a gap between the upper and lower regret bounds concerning the dimension $d$ of the feature vectors, it is due to the inheritance of the approximation ratio of the G-optimal design with a barycentric spanner from existing studies.

**Strengths:**

- The paper proposes a novel and impactful algorithm.
- The design and motivation of the proposed algorithm are clearly explained.
- A well-summarized proof sketch is provided in the main body.

**Weaknesses:**

- The regret upper bound of the proposed algorithm is not optimal for the dimension $d$ of the feature vectors.

**Questions:**

#### Questions
- No specific questions.

#### Comments
- Overall, I believe this is a high-quality paper.
- In my opinion, this paper should include in the main body the choice of batch size and how it is derived (or citations) if space allows.

#### Minor Comments
- Line 201: an uniform -> a uniform
- Line 287: $B_1$ -> $B_{1/T}$

**Limitations:**

- In this paper, limitations are well discussed (e.g., assumptions for the problem).
- While societal impacts are not discussed in this paper, it could be considered a paper with little direct impact on society since it is a theoretical paper.

---

> ### Author Rebuttal · Authors · 2023-08-10
>
> We thank the reviewer for their input and for appreciating our contribution.
> * **[Regret dependency on $d$]** We would like to highlight that our regret bounds match the best known regret bounds for state-of-the-art (unbatched) efficient contextual linear bandit algorithms [Agrawal et al., 2013; Zhu et al., 2022; Dani et al., 2008].\
> \
> Our worst-case regret bound loses a $\sqrt{d}$ factor as compared to the $\Omega(d\sqrt{T})$ lower bound, however it is not clear that $\tilde{O}(d\sqrt{T})$ regret bound is achievable with an oracle-efficient algorithm.\
> \
> The reason we lose this factor is as follows.
> In our algorithm, a critical step is to find a small subset of actions (spanner actions) that form good bases of the entire action space.
> We use linear optimization oracles to achieve this computationally efficiently.
> However, this method can only produce spanner actions with a $V^{-1}$-norm (defined in (5) of the paper) bounded by $O(\sqrt{d})$, which gives the corresponding factor in the regret bound. Any future improvement that reduces the norm from $O(\sqrt{d})$  to $O(1)$ will immediately result in nearly optimal regret bounds for our algorithms.
> * **[Writing]** Thank you for pointing out these typos. We have corrected these in our updated version.

---

> > ### Comment · Reviewer_DZuy · 2023-08-18
> >
> > Thank you for your response. I have read the response and other reviews.
> > I think the dependency on $d$ is a minor point, and the possibility of improvement is well discussed in the paper.
> > I keep my score.

---

### Official Review · Reviewer_azE5 · 2023-07-06

**Soundness:** 3 good
**Presentation:** 3 good
**Contribution:** 3 good
**Rating:** 6
**Confidence:** 3

**Summary:**

This paper proposes a new batched contextual linear bandits algorithm that achieves $\widehat{O}(\sqrt{T})$ regret while using $O(\log\log T)$ batches. The algorithm combines various techniques from previous work including approximate barycentric spanner computation and reduction from contextual linear bandits to linear bandits. Meanwhile, the algorithm is also computationally efficiently.

**Strengths:**

- The algorithm intuition and application of prior techniques of approximate barycentric spanner computation and reduction from contextual linear bandits to linear bandits are considered to be novel.
- The algorithm is shown to be computationally efficient while maintaining moderately good regret and low number of batches.

**Weaknesses:**

Although it is proven that the proposed algorithm has good computational complexity, it will be better if we can empirically see its computational efficiency is superior over other algorithms listed in Table 1. In particular, it seems that implementing the contextual version of the algorithm (Algorithm 3) is not very trivial. Can we see some empirical comparisons of regret and computational efficiency among Algorithm 3 and some other algorithms listed in Table 1?

### Suggestions on Writing
- It can be better if specific lemmas are referred when the arguments in main context use any results from the appendix. For example, the arguments in line 248 and 277.
- It may be better to define $\Delta(a)$ clearly after defining $L_z(a)$ in line 273.
- For self-containedness, it will be better to have a brief explanation of how Algorithm 2 works.

**Questions:**

- Why the regret bound based on the minimum gap is not applicable for contextual linear bandits?
- A better G-optimal design approximation can be obtained when $\mathcal{A}_t$ is discrete and finite. Will this give better theoretical guarantee?

**Limitations:**

The theoretical limitations are well-addressed in the paper.

---

> ### Author Rebuttal · Authors · 2023-08-10
>
> We thank the reviewer for their input and for appreciating our contribution.
>
> * **[Empirical results]** Indeed, as the reviewer notes, in this work we focused on the theoretical understanding of efficient batch learning in contextual bandits, where we show a batched bandit algorithm that provably achieves computational efficiency while maintaining near-optimal regret bound.
> To illustrate this, let us consider a $d$-dimensional linear bandit problem with $T^d$ actions.
> Our algorithm takes only $\mathrm{poly}(dT)$ time per batch whereas previous batched algorithms can take $\Omega(T^d)$ time, exponential in $d$.\
> \
> Having said that, following the reviewer's suggestion, we have done a small experiment to compare the computational complexity of computing the exploration policy of our algorithm versus the complexity of computing the policy in [Zhang et al., 2021] (complexity of one batch). We do not consider other batched algorithms such as [Ruan et al., 2021] since they are not feasible to implement even for a small number of arms. We used $d=5$ dimensions and a batch of size $100$ iterations. For simplicity we use a fixed action set (unit sphere), however, this knowledge is not revealed to the algorithms, i.e., the algorithms assume that the action set may change over time.
> As the policy of [Zhang et al., 2021] requires to solve a non-convex optimization problem, it is not feasible to implement it for infinite number of arms. Instead, we solve the optimization problem over a finite subset of $k$ arms sampled uniformly at random from the action set. In contrast, our algorithm can be directly applied for the infinite action set, hence, the computational complexity will not depend on $k$. In the figure attached to the common rebuttal, we plot the time complexity versus the sampled number of arms (on Intel(R) Xeon(R) CPU @ 2.20GHz, 56MB cache). We observe that for moderately large number of arms, our algorithm achieves significant savings in computational complexity as compared to the scheme of [Zhang et al., 2021].\
> \
> We note again that theoretically we have proved the same order regret bounds; but we will be happy to add numerical evaluation in the next version.
> * **[Writing]** Thank you. We will incorporate the suggested changes above, as well as the response to the questions of the reviewer in the next version.
> * **[Gap-dependent regret bounds for contextual linear bandits]** Thank you for this very good question. The main reason is that a large minimum action gap in the original action sets $A_t$ does not imply a large gap in the reduced action set $X$. As a simple example consider $d=1$, $\theta_\star = 1$, and two action sets $A_1=\\{-1,1\\}$, and $A_2=\\{-1\\}$. At each iteration the learner receives the action set $A_1$ with probability $p$ and $A_2$ with probability $1-p$ independently from other iterations. Recall that the action set in the reduced instance $X=\\{g(\theta)|\theta \in [-1,1]\\}$, where $g(\theta)= E_{A\sim D}[\arg\max_{a\in A} a^\top \theta]$. For $\theta \geq 0$ we have that $g(\theta)=p(1)+(1-p)(-1)=2p-1$, while for $\theta < 0$ we have $g(\theta)=p(-1)+(1-p)(-1)=-1$. Then $X=\\{-1,2p-1\\}$. Therefore, the suboptimality gap is $\Delta_{\min}=(2p-1)(1)-(-1)(1)=2p$ which can be arbitrarily small depending on $p$. Note that in the original contextual bandit instance, the minimum gap is at least $2$ for both action sets.\
> \
> While it may be possible to provide gap dependent regret bounds for our algorithm in the contextual case, this will require more sophisticated regret analysis that does not solely rely on the reduced linear bandit instance.
> We are happy to include this discussion in the next version of our paper.
> * **[Discrete and finite sets of actions]** Indeed, a better G-optimal design will directly translate to better regret bounds for our algorithms. However, the analysis may become  more involved in the contextual case. Moreover, to the best of our knowledge, these results are obtained for small action sets while we focus in this work on large action spaces.

---

> > ### Comment · Reviewer_azE5 · 2023-08-10
> > **Response**
> >
> > Thank you very much for your rebuttal and my concerns have been well-addressed.

---

> > > ### Author Response · Authors · 2023-08-17
> > >
> > > Thank you for your response and the helpful comments. Please let us know if you have any additional questions or concerns.

---

### Official Review · Reviewer_vkke · 2023-07-12

**Soundness:** 2 fair
**Presentation:** 3 good
**Contribution:** 2 fair
**Rating:** 6
**Confidence:** 3

**Summary:**

This paper studies batched online learning for linear bandits and linear contextual bandits in large action space. It shows that the learner can use $O(\log \log T)$ batches (in each of which the sampling distribution over actions remain the same) to achieve near-optimal regret. Compared to previous work, the main contribution is to introduce an oracle efficient algorithm which only requires linear optimization over the action set, and thus can handle large action space. The regret bound is a $d$ factor worse than the tight bound.

**Strengths:**

- The paper studies an important question for linear (contextual) bandits where the action space is large and the number of policy switch is limited. The proposed algorithm is natural and easy to understand. The regret bound (just a $d$ factor sub-optimal) is the best we can hope for under the computational constraint.
- There are some other algorithmic ideas developed, which could be of independent interest. For example, adding a little ball around the origin to make the action space full rank, and the "action set shaping" technique that balance exploitation and exploration when finding the solution of the approximate G-optimal design.

**Weaknesses:**

- Though I'm not entirely familiar with the literature, I feel that the contribution might be a bit incremental. The reason is that in batched online learning, ways to achieve optimal performance using $O(\log \log T)$ batches has already been established. The main contribution of this work is about using barycentric spanner to efficiently explore. The related technique has been extensively studied in [Zhu et al.], though not for the batched setting. I can see a lot of similarity between the LW-ArgMax in this paper and the IGW-ArgMax in [Zhu et al.], but no technical comparison is provided. The LWS procedure is also closely related to the ReweightedSpanner procedure in [Zhu et al.]. To be fair, the two are still slightly different because this paper uses inverse square gap weighting while theirs is inverse gap weighting, but I think there still needs to be some comparison.

- The proposed algorithm operate on the extended action set $\mathcal{A}'$, which seems to require a linear optimization oracle over $\mathcal{A}'$. I don't think it is explained what if the learner only has an oracle for the original action set $\mathcal{A}$. It is unclear whether an oracle for $\mathcal{A}'$ can be straightforwardly constructed from that for $\mathcal{A}$.

- Similarly, in the contextual setting, the algorithm operates over a very artificial action set $g^{(m)}([\theta]_q)$. It is totally unclear how to leverage the linear optimization oracle for the original action set $\mathcal{A}_t$ to implement the algorithm.

- Since a inverse square gap weighting is used, maybe it's possible to show an refined bound like in [Lattimore and Szepesvari, 2016]. Now only a $\Delta_{\min}$ dependent instance-dependent bound is shown, which can be highly sub-optimal.


[Zhu et al., 2022] Zhu, Foster, Langford, Mineiro. Contextual bandits with large action spaces: made practical.

[Lattimore and Szepesvari, 2016] The End of Optimism? An Asymptotic Analysis of Finite-Armed Linear Bandits

**Questions:**

See the weakness section.

**Limitations:**

There is no societal impact.

---

> ### Author Rebuttal · Authors · 2023-08-10
>
> We thank the reviewer for their review and valuable comments.
>
> * **[Distinction from existing algorithms]** We respectfully disagree that the contributions in our work are incremental with respect to previous work. Our work provides the first efficient batched algorithms, both for linear and contextual bandits, that we believe can be of significant practical interest, and that required the development of new algorithms and proof techniques for their analysis.\
> \
> While batched algorithms for contextual linear bandits that use $O(\log \log T)$ batches indeed exist [Ruan et al., 2021], these algorithms are not computationally-efficient, and it is not a direct extension making them so. These algorithms use a G-optimal design, as we also do. However, while using a Barycentric spanner to get the G-optimal design is already used in the literature [Awerbuch et al., 2004], this cannot make existing algorithms computationally efficient. The reason is that, these algorithms require a G-optimal design over a good set of actions (since they use action elimination) that is represented by non-convex constraints. Hence, finding a Barycentric spanner requires solving a non-convex (non-trivial) optimization problem. To deal with this, our paper proposes a novel algorithm that does not use "hard arm elimination" (Algorithm 1); instead, we use a form of "soft arm elimination" through inverse squared gap weighting. Due to the less frequent updates of the estimated gaps (only $O(\log \log T)$ times), analyzing the effect of the inverse squared gap weighting on the concentration of estimated action means becomes challenging. Due to space limitations, we are happy to provide more details about this in the comments if the reviewer is interested. We next provide more details in distinction from specific works.
> * **[Distinction from [Zhu et al., 2022]]** While constructing a linear optimization oracle with inverse gap weighting is a related problem, as we also mention in footnote 7 of our submission, the resulting strategy (Algorithm IGW-ArgMax in [Zhu et al., 2022]) is not applicable when inverse squared gap weighting is used. Our technique and the scheme of [Zhu et al., 2022] both use the standard idea of line search; the technical challenge is how to construct the function and step of the line search. In particular, for inverse gap weighting calling the optimization oracle by doing a line search with $N=O(\log T)$ steps for the function $z \theta + z^2.C. \Delta(a)$ provides an output with sufficiently small approximation error. However, for inverse squared gap weighting we do a line search with $N=O(\log^2 T)$ steps for the function $z (1+1/W)\theta + z^{1+1/W}.C. \Delta(a)$ with $W=3\log T$. The proof that this line search provides an approximate optimization oracle is much more involved than the inverse gap weighting case (please see App. B for details of the proof). We are happy to add a detailed technical comparison in our appendices.
> * **[LWS vs ReweightedSpanner of [Zhu et al., 2022]]** Indeed, the LWS algorithm and ReweightedSpanner in [Zhu et al., 2022] are similar since this is a standard method, proposed in [Awerbuch et al., 2004], to construct a Barycentric spanner using a linear optimzation oracle. The main challenges are: (i) in designing an algorithm that requires Barycentric spanners over sets that admit an efficient optimization oracle, and (ii) constructing a linear optimization oracle to use in the LWS algorithm.\
> \
> In addition to the distinctions above, the work in [Zhu et al., 2022] does not face the challenge of less frequent updates of the estimated gaps due to the batch constraints. As a result, the regret analysis of our algorithms and [Zhu et al., 2022] use very different techniques. The less frequent updates of the estimated gaps also pose a difficulty in directly analyzing the regret for contextual bandits. This is the main reason that, unlike [Zhu et al., 2022], our analysis is done for linear bandits and carried out to the contextual bandit case using the reduction in [Hanna et al., 2022].\
> \
> In summary, our proposed algorithm, method of constructing a linear optimization oracle over the weighted action set, and regret analysis cannot be easily obtained from existing work in the literature.
>
> * **[Optimization oracle over $A'$]** Since $A'=A\cup B_{1/T}$, the linear optimization problem $\max_{a\in A'}\langle a, \theta \rangle $ can be solved by comparing $\max_{a\in A}\langle a, \theta \rangle $ and $\max_{a\in B_{1/T}}\langle a, \theta \rangle = 1/T$ (note that $\arg\max_{a\in B_{1/T}}\langle a, \theta \rangle = (1/T) \theta /{\|\theta\|_2}$). Indeed, in our paper we only assume an optimization oracle over the original action set $A$. We are happy to clarify this in the next version.
>
> * **[Optimization oracle over the reduced set $X_m$]** We do provide an efficient optimization oracle over the artificial action set $X_m$. As mentioned in page 9 (line 343), the function $g^{(m)}([\theta]_q)$ for $q\leq \epsilon/2$ serves as our (approximate) linear optimization oracle. This function can be computed by calling the linear optimization oracle for the original action set $A_t$ at most $T$ times. The formal proof is provided in App. F.
>
> * **[Tighter instance dependent regret bounds]** We would like to highlight that ours is the first result to achieve nearly optimal gap dependent regret of $\tilde{O}(1/\Delta_{\min})$ using an efficient batched algorithm. General instance-dependent bounds can be a future direction of our work. [Lattimore and Szepesvari, 2016] considers the asymptotic regret analysis, while in our submission we focus on the finite time regret analysis. It might be possible for us to get similar bounds as in [Lattimore and Szepesvari, 2016], but we leave it as a future work as well.

---

> > ### Comment · Reviewer_vkke · 2023-08-16
> >
> > I thank the authors for addressing several of my concerns. I have raised the score.

---

> > > ### Author Response · Authors · 2023-08-17
> > >
> > > Thank you so much for increasing the score and for taking the time to review our response. Please let us know if you have any additional questions or concerns.

---

### Author Rebuttal · Authors · 2023-08-10

We thank all the reviewer for their input and for appreciating our contribution.  This common part is devoted to summarizing the novelty of our proposed schemes. Detailed responses on the reviewers questions  are provided as a response to each reviewer. We also attach a figure with preliminary simulation results to reference in our comments.
* **[Novelty in the reduction (Question from Reviewer 6gSX)]**  Section 4 of our submission is devoted to establishing an efficient batched algorithm for contextual bandits. The core idea is to leverage the oracle-efficient batched linear-bandit algorithm in Section 3 and the reduction ideas established in [18]. However, the reduction in [18] is not focusing on time efficiency and a straightforward application does not result in a desirable (in terms of efficiency) algorithm. Moreover, when the context distribution is unknown, the reduction in [18] uses different batch lengths to estimate the reduction function $g$ -- this makes it incompatible with our batched linear bandit algorithm. The core novelty in this section lies in the solution of the above two issues.\
\
Indeed, to resolve the incompatible batch length problem,  we tune the batch lengths to have sufficiently good estimates of $g$ while only increasing the number of batches by a factor of $2$ as compared to the linear bandit case.\
\
However, solving the efficiency issue requires much more effort. In particular, we establish an efficient approximate linear optimization oracle on the reduced action set with the optimization oracle over the original context.
This new optimization oracle can then be applied with Algorithm 1 to solve the reduced linear bandit instance. Note that this is highly non-trivial as the size of the reduced action set can grow as $T^{Cd}$ for some constant $C$. Additional computation issues arise from the computation of the inverse of the reduction function $g$ (defined in line 321) as required in the reduction in [18]. We provide an efficient solution to it with an approximate optimization oracle constructed by the original optimization oracle (see Assumption 2).



* **[Novelty in bounding the regret (Question from Reviewer 6gSX)]** We here provide a summary of the novelty in our approach  in Section 3.2. The core idea of the algorithms consists of two steps. For each batch, $m$, the first step is identify a small set of actions $C_m$ (a barycentric spanner) so that pulling these actions can give a better estimator of the parameter $\theta^*$ than the previous batch;
the second step is to ensure that for each arm $a\in C_m$, it is pulled for a number of times  $\propto 1/\Delta_a^2$, where  $\Delta_a=(a_\star-a)^\top \theta_\star$ is the gap for action $a$.
As such, the overall regret for this batch is at most $\propto \sum_{a\in C_m} 1/\Delta_a$, a form close to the optimal regret formulation. Hence, if these two steps are possible, then the overall algorithm would have a small regret. However, the above steps appear to be conflicting to each other. For instance, if we use the existing approach to establish barycentric spanners on the action set, then pulling each $a$ a number of times $\propto 1/\Delta_a^2$ does not give a good estimator to $\theta^*$ (the existing barycentric spanner proof actually requires that each $a$ needs to be pulled a equal number of times, regardless of the gaps). We resolve this by performing a careful weighting on the actions and find a barycentric spanner on the weighted actions. Then we show that pulling $\propto 1/\Delta_a^2$ times on the barycentric spanner of the weighted actions can give a good estimator -- resolving the conflicts.\
\
We also would like to highlight that the weighting on the actions are also highly non-trivial as the weights depend on the gaps of the actions, which are unknown in the first place. Our approach cleverly avoids the estimation of the gaps for all actions but only these actions that are called by the optimization oracle.

---

### Decision · Program_Chairs · 2023-09-21

**Decision:**

Accept (poster)

**Comment:**

This paper considers the linear contextual bandit setting. The main novelty of this paper is achieving a sqrt(d^3 T) regret bound using just loglog(T) rounds of adaptivity. This is an improvement of the log(T) rounds that exists in the literature. The regret bound does not match the minimax optimal rate of sqrt(d^2 T), but the authors point out that this may be the price of computational efficiency, as prior works also achieve this rate using the argmax oracle. In summary, achieving a regret bound that scales as sqrt(T) with just loglog(T) rounds of adaptivity and being computational efficient is a significant advance. As reviewers, we also thank the authors for the experimental evaluation given in the rebuttal phase, despite this being a theoretical paper.